# Distributionally Robust Multi-Agent Reinforcement Learning for Dynamic Chute Mapping

**Guangyi Liu** [1]  **Suzan Iloglu** [1]  **Michael Caldara** [1]  **Joseph W. Durham** [1]  **Michael M. Zavlanos** [1 2]

## Abstract

In Amazon robotic warehouses, the destination-to-chute mapping problem is crucial for efficient package sorting. Often, however, this problem is complicated by uncertain and dynamic package induction rates, which can lead to increased package recirculation. To tackle this challenge, we introduce a Distributionally Robust Multi-Agent Reinforcement Learning (DRMARL) framework that learns a destination-to-chute mapping policy that is resilient to adversarial variations in induction rates. Specifically, DRMARL relies on group distributionally robust optimization (DRO) to learn a policy that performs well not only on average but also on each individual subpopulation of induction rates within the group that capture, for example, different seasonality or operation modes of the system. This approach is then combined with a novel contextual bandit-based estimator of the worst-case induction distribution for each state-action pair, significantly reducing the cost of exploration and thereby increasing the learning efficiency and scalability of our framework. Extensive simulations demonstrate that DRMARL achieves robust chute mapping in the presence of varying induction distributions, reducing package recirculation by an average of 80% in the simulation scenario.

## 1. Introduction

In Amazon robotic sortation warehouses, mobile robots are deployed to transport and sort packages efficiently to different destinations (Wurman et al., 2008; Azadeh et al., 2019; Amazon, 2022b;a; 2023). The sorting process be-

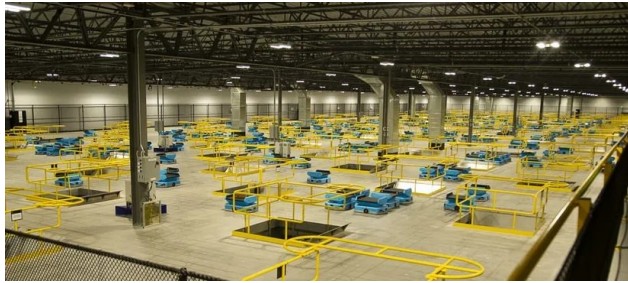

Figure 1: Schematic layout of an Amazon robotic sortation warehouse featuring robot drives and eject chutes.

gins at induction stations, where packages are loaded onto mobile robots and subsequently transported to designated eject chutes based on their destinations (Figure 1). A critical factor determining the package throughput capacity of these facilities is the effective allocation of eject chutes to different destinations. Therefore, the destination-to-chute mapping policy plays a crucial role in optimizing the overall throughput performance of the robotic sortation warehouses.

Our previous work (Shen et al., 2023) addresses the destination assignment problem (DAP) (Boysen & Fliedner, 2010) in robotic sortation systems by developing a dynamic chute mapping policy. This policy determines the optimal allocation of eject chutes to destinations with the objective of minimizing the number of unsorted packages. We proposed a model-free reinforcement learning approach that dynamically adjusts the number of chutes assigned to each destination throughout the day. Our solution formulates the chute mapping problem within a Multi-Agent Reinforcement Learning (MARL) framework (Lowe et al., 2017; Sunehag et al., 2017; Samvelyan et al., 2019; Rashid et al., 2020), where each destination is represented as an agent that controls its chute allocation at each time step.

While the MARL policy proposed in our previous work (Shen et al., 2023) demonstrates superior performance compared to traditional reactive chute mapping approaches often implemented in Amazon robotic sortation warehouses, its effectiveness assumes that the induction distribution during deployment matches the training distribution and that the daily induction rate remains close to its average value.

[1]Amazon Robotics, North Reading, MA, USA [2]Department of Mechanical Engineering and Materials Science, Duke University, Durham, NC, USA. Correspondence to: Guangyi Liu <gyliu@amazon.com>.

*Proceedings of the 42nd International Conference on Machine Learning*, Vancouver, Canada. PMLR 267, 2025. Copyright 2025 by the author(s).

In practice, however, induction patterns exhibit significant temporal variations, potentially compromising the MARL policy's performance when confronted with unexpected distribution changes.

To enhance robustness against such variations, in this paper, we propose a Distributionally Robust Multi-Agent Reinforcement Learning (DRMARL) framework that learns chute mapping policies capable of maintaining near-optimal performance across diverse induction distributions. Specifically, we formulate this problem as group DRO problem, where each group represents a distinct induction distribution pattern extracted from a subset of a historical dataset. Our DRMARL framework optimizes policies for the worst-case induction patterns across these distribution groups. To address the computational cost of exhaustively evaluating all distribution groups during training, we introduce a contextual bandit (CB)-based worst-case reward estimator for each state-action pair. Through extensive evaluation, we demonstrate that our DRMARL framework yields robust chute mapping policies that not only outperform baseline MARL policies on out-of-distribution (OOD) induction data but also maintain consistent performance across varying induction distributions.

Our *contributions* are twofold: First, we introduce group distributionally robust optimization in multi-agent reinforcement learning, developing a novel framework to learn policies that are robust to distribution shifts in the reward function. Second, we propose an novel contextual bandit-based method for efficient worst-case reward estimation, significantly reducing the computational complexity of DRMARL training by eliminating the need for exhaustive group exploration to estimate the worst-case reward. To the best of our knowledge, our framework is the first to integrate contextual bandits with group DRO and MARL, addressing a well-known challenge of distributionally robust reinforcement learning related to its computational cost. Our proposed framework has broad applicability to various large-scale industrial applications beyond sortation systems, including resource allocation, collaborative robotics, and warehouse automation, where robustness to distribution shifts is crucial.

### 1.1. Literature Review

*Destination Assignment Problems*: Mathematical programming has been used to optimize warehouse systems, including destination assignment problems (DAP) (Boysen & Fliedner, 2010) for sorting systems. The destination mapping approach in (Fedtke & Boysen, 2017) optimizes package flow by minimizing travel distances between inbound and outbound stations in conveyor-based sorting systems, which leads to improved throughput. (Novoa et al., 2018) minimizes the worst-case flow imbalance across work sta-

tions on the sortation floor, developing a stochastic approach with chance and robust constraints. For robotic sorting systems, (Khir et al., 2021) proposes an integer programming method to solve DAPs that minimize sortation effort and satisfy package deadlines. A robust formulation addressing demand uncertainty is presented in (Khir et al., 2022). While these approaches effectively optimize destination assignment in sorting systems, they do not account for distributional uncertainty in demand and system dynamics. In contrast, our proposed DRMARL framework explicitly models such uncertainties, ensuring robust performance under varying operational conditions.

*MARL for Resource Allocation*: MARL has previously been applied to address resource allocation problems (Nie et al., 2021; Mei & Wang, 2024; Jun-Han et al., 2025). For example, a MARL framework for ocean transportation networks was proposed in (Li et al., 2019). This framework developed a multi-agent Q-learning algorithm where the local Q-networks depend on the joint states (including the limited shared resources) and the joint actions. However, since the joint state-action space grows exponentially with the number of agents, the local Q-networks are hard to learn and this approach does not scale well in practice. This limitation was addressed in our previous work (Shen et al., 2023), where the local Q-networks are only loosely coupled, enhancing the scalability while still being interconnected enough to capture the impact of robot congestion on the sortation floor. Compared to (Li et al., 2019), the method proposed in (Shen et al., 2023) models resources explicitly as actions and considers budget constraints when taking joint actions. However, these MARL-based approaches do not incorporate distributional robustness, making them sensitive to demand fluctuations and uncertainty, which our DRMARL framework explicitly addresses to ensure reliable performance in dynamic sorting environments.

*Robust and Distributionally Robust RL*: Robust Reinforcement Learning (Robust RL) (Morimoto & Doya, 2005; Wiesemann et al., 2013; Pinto et al., 2017; Panaganti & Kalathil, 2021; Moos et al., 2022; Panaganti et al., 2022; Goyal & Grand-Clement, 2023; Yamagata & Santos-Rodriguez, 2024) develops policies that maintain performance under worst-case conditions through adversarial perturbations. Distributionally Robust Reinforcement Learning (DRRL) (Xu & Mannor, 2010; Smirnova et al., 2019; Hou et al., 2020; Wang et al., 2023; Ramesh et al., 2024; Zhang et al., 2024; Lu et al., 2024; Panaganti et al., 2024) extends this by optimizing across environment distributions rather than a single worst-case scenario. While traditional DRRL primarily addresses ambiguity in MDP transition probabilities, this approach inadequately captures induction distribution changes in Amazon robotic sortation warehouses. Our problem requires a focus on distributionally robust optimiza-

tion of reward function distributions, building on (Ren & Majumdar, 2022; Liu et al., 2022). Recent advances in (Distributionally) Robust Multi-Agent RL (Zhang et al., 2020; Bukharin et al., 2024; Shi et al., 2024b;a) have introduced frameworks like RMGs, ERNIE, and DRNVI to address environmental uncertainties, adversarial dynamics, and model uncertainties. While existing methods primarily focus on robustness in transition dynamics, adversarial interactions, and general environmental uncertainties, they do not explicitly address distributional shifts in package induction, which is a critical challenge in sortation warehouses. Our approach extends DRMARL to explicitly model and optimize against uncertainties in induction distributions, ensuring robust and consistent performance under varying operational conditions.

*Group DRO:* Group Distributionally Robust Optimization aims to enhance model robustness across diverse subpopulations by optimizing for the worst-performing groups rather than the average performance (Sagawa et al., 2020). This approach ensures fairness and resilience to distribution shifts, particularly for underrepresented groups. While initial work focused on single-agent supervised learning (Hu et al., 2018; Oren et al., 2019), recent advances have extended these principles to more complex settings. Notably, (Soma et al., 2022) proposed a soft-weighting method on distribution groups with convergence guarantees, while (Wu & Fu, 2023) and (Xu et al., 2023) demonstrated the applicability of group DRO in multi-agent systems and reinforcement learning, respectively. Our work bridges a critical gap by introducing group DRO principles to DRMARL. We begin by formulating the distributionally robust Bellman operator and addressing the computational challenges of exploring all distribution groups during training. To tackle these challenges, we provide a DR Bellman operator specifically designed for MARL and introduce a contextual bandit (CB)-based worst-case distribution group estimator. This estimator enables efficient training by adaptively identifying the worst-case distribution groups.

The remainder of the paper is organized as follows. In Section 2, we formulate the dynamic chute mapping problem within a multi-agent reinforcement learning framework. In Section 3, we extend this formulation by incorporating group distributionally robust optimization into MARL, while in Section 4, we present novel contextual bandit-based worst-case reward estimator to enhance training efficiency. Finally, in Section 5, we demonstrate the effectiveness of our proposed framework through extensive simulations.

## 2. Problem Formulation

In Amazon robotic sortation warehouses, package flow is modeled using three buffers: induct, laden drive, and recirculation, as illustrated in Figure 2. The laden drive buffer

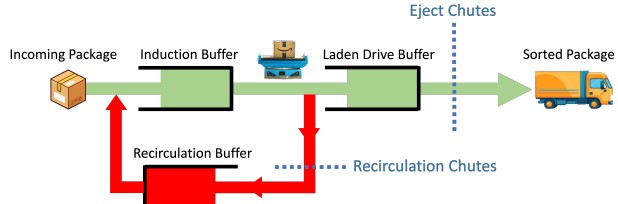

Figure 2: Flow of packages in the Amazon robotic sortation warehouse.

contains packages actively transported by robots from induct stations to assigned eject chutes. Since the total number of chutes is limited, contention occurs when multiple destinations require chute capacity simultaneously. When chute availability is insufficient, excess packages are diverted to recirculation chutes, entering the recirculation buffer for another pass through the system. Chutes are reallocated to different destinations when a destination vehicle reaches capacity or when its scheduled departure time is reached. The dynamic chute mapping policy optimizes chute allocations across destinations in real time to minimize recirculation and maximize system throughput.

We formulate the dynamic chute mapping problem as a sequential decision-making problem, specifically as a multi-agent reinforcement learning (MARL) problem that determines the optimal chute allocations to minimize package recirculation at each time step. To this end, we define a Markov game over $N$ unique destinations, represented by the tuple $\left(N, \mathcal{S}, \{\mathcal{O}^i\}_{i=1}^N, \{\mathcal{A}^i\}_{i=1}^N, P, \{r^i\}_{i=1}^N, \gamma, \rho_0\right)$, where $\mathcal{S}$ is the joint state space, $\mathcal{O}^i \subset \mathcal{S}$ is the local observation space for destination $i$, $\mathcal{A}^i$ denotes the action space (i.e., the number of new chutes acquired) for destination $i$, $P$ is the state transition probability, $r^i$ is the reward function, $\gamma \in (0, 1)$ is the discount factor, and $\rho_0$ is the initial state distribution. While the joint state provides complete system information, each destination's observation is limited to local features relevant to its decision-making. This partial observability requires destinations to act based on their own experiences and available information. We refer to Appendix C.1 for a detailed MARL formulation.

The system operates in discrete time steps, where, at each step $t$, individual agents (destinations) make chute allocation decisions. Each agent $i$ employs a local policy $\pi^i : \mathcal{O}^i \times \mathcal{A}^i \rightarrow [0, 1]$ that maps local observations $o^i$ to probabilities over possible chute allocation actions. Each agent $i$ learns its optimal local policy $\pi^{i,*}$ by maximizing the expected discounted return $\mathbb{E}[R_t^i] = \mathbb{E}\left[\sum_{t'=t}^{\infty} \gamma^{t'-t} r_{t'}^i\right]$, where $r_{t'}^i$ denotes the instantaneous reward at time $t'$ and $\gamma$ is the discount factor. The expectation captures both the stochastic nature of the policy and the environment dynamics. This formulation naturally aligns individual agent objectives with the global goal of minimizing recirculation

while maintaining the throughput. The instantaneous reward function for agent $i$ at time step $t$ is defined as:

$$r_t^i = -\texttt{recirc}_t^i - 2\,a_t^i, \tag{1}$$

where $\texttt{recirc}_t^i \geq 0$ represents the number of packages in recirculation for destination $i$ and the action term $a_t^i$ is included as penalty to prevent the learned policy from over-allocating chutes to only a few destinations. Due to the coupled nature of agent decisions, we utilize the joint action-value function to determine optimal local policies $\pi^{i,*}$:

$$Q^\pi(s,a) = \mathbb{E}\Big[\sum_{i=1}^{N} R_t^i | s_t = s, a_t = a\Big], \tag{2}$$

which evaluates the expected return when taking the joint action $a = (a^1, \ldots, a^N)$ in state $s$ and following the joint policy $\pi$ thereafter.

To mitigate the exponentially growing policy space, we assume agents execute actions independently, such that $\pi = \prod_{i=1}^{N} \pi^i$. The optimal policy $\pi^*$ is learned using the Deep Q-Network (DQN) (Mnih et al., 2015), where a neural network $Q(s, a; \theta)$ with parameters $\theta$ approximates the optimal action-value function $Q^{\pi^*}(s, a)$. The learning process minimizes the loss:

$$L(\theta; X) = \mathbb{E}_{s,a,r,s'}\left[(Q(s,a;\theta) - y)^2\right], \tag{3}$$

where $y = \mathbb{E}_{X \sim \mathbb{P}}[r(s, a; X)] + \gamma \max_{a'} \bar{Q}(s', a'; \bar{\theta})$ approximates the optimal target values $\mathbb{E}_{X \sim \mathbb{P}}[r(s, a; X)] + \gamma \max_{a'} Q^{\pi^*}(s', a')$. Here, $r(s, a; X)$ represents the instantaneous reward under the current state-action pair and induction distribution $X$[1]. Stability of the learning process is enhanced through two mechanisms: a target network $\bar{Q}$ with periodic parameter updates using the most recent values of $\theta$, and an experience replay buffer $\mathcal{D}$ that stores transition tuples $(s, a, r, s')$. The resulting optimal policy takes the form:

$$\pi^*(s,a) = \begin{cases} \frac{1}{|\mathcal{A}(s)|} & \text{if } a \in \mathcal{A}(s), \\ 0 & \text{otherwise} \end{cases} \tag{4}$$

where $\mathcal{A}(s) = \arg\max_a Q(s, a; \theta^*)$ and $\theta^* = \arg\min L(\theta; X)$.

To address the scalability of the state-action space and the computational feasibility of the expectation in (3), we employ the Value Decomposition Network (VDN) (Shen et al., 2023) with budget constraints for computing feasible joint actions, with implementation details provided in Appendix C.3 and Appendix C.4.

While the above MARL-based chute mapping policy demonstrates strong performance under standard operating conditions, it exhibits significant performance degradation under

---

[1]See Appendix C.2 for details of the induction distribution.

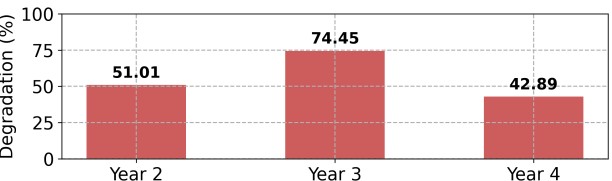

Figure 3: Relative degradation of MARL policy (trained on Year 1) on OOD (Years 2-4) induction data, compared to Year 1.

distribution shifts in package induction patterns (see Figure 3). This vulnerability to out-of-distribution scenarios motivates our robust formulation.

In this paper, our *objective is to introduce robustness to distribution shifts* in the learned chute mapping policies. We achieve this by incorporating group DRO into the MARL framework, giving rise to the proposed DRMARL approach. Our proposed framework ensures reliable and robust performance in Amazon robotic sortation warehouses, even under unforeseen future induction scenarios.

## 3. Distributionally Robust Multi-Agent Reinforcement Learning with Group DRO

In this section, we enhance the MARL chute mapping framework described in Section 2 by incorporating group DRO to handle uncertainty and variability in package induction distributions. This approach enables us to develop robust policies that perform well across diverse induction scenarios, including previously unseen induction distributions.

Unlike traditional stochastic optimization, which assumes that the true data-generating distribution is known and fixed, Distributionally Robust Optimization (DRO) addresses uncertainty in the underlying distribution by optimizing against a family of plausible distributions, i.e., the ambiguity set, constructed from the available data (Delage & Ye, 2010; Shapiro, 2017; Rahimian & Mehrotra, 2022; Zhen et al., 2023; Liu et al., 2024; Kuhn et al., 2024; Konti et al., 2024). Instead of minimizing the expected loss under a single estimated distribution, DRO seeks solutions that minimize the worst-case expected loss over all distributions within this ambiguity set. This leads to policies that are robust to distributional shifts, such as those encountered in the chute mapping problem, where test-time conditions may deviate from those seen during training.

### 3.1. Group DRO

In DRO, the ambiguity set that captures uncertainty in the data-generating distribution can be defined in various ways. In particular, Group DRO (Hu et al., 2018; Oren et al., 2019) offers an efficient way to define the ambiguity set using a finite collection of known distributions. In the context of

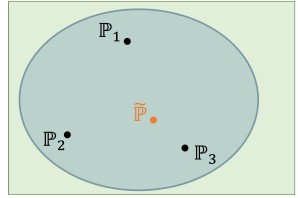 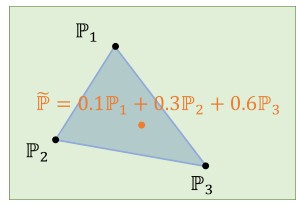

Figure 4: Ambiguity set $\mathfrak{M}$ in regular DRO (Rahimian & Mehrotra, 2022) (left) versus group DRO (Sagawa et al., 2020) (right).

robotic sortation warehouses, this approach is especially useful, as the ambiguity set can be constructed from historical induction data that reflects the variability observed in real-world operations.

Following (Sagawa et al., 2020), we define the unknown distribution $\tilde{\mathbb{P}}$ as a combination of $m$ distributions $\mathbb{P}_g$, each indexed by a group $g$ in the set $\mathcal{G} = \{1, 2, \ldots, m\}$. The ambiguity set $\mathfrak{M}$ is then defined as a convex combination of these groups:

$$\mathfrak{M} := \left\{ \tilde{\mathbb{P}} = \sum_{g=1}^{m} q_g \, \mathbb{P}_g \ \Big| \ q \in \Delta_m \right\}, \tag{5}$$

where $\Delta_m$ denotes the $(m-1)$-dimensional probability simplex (Grünbaum et al., 1967) (see Figure 4).

In the dynamic chute mapping problem, we assume that past years of operational data from sortation warehouses provide sufficiently rich historical induction distributions that can be used to obtain representative distribution groups $\mathcal{G}$. With this assumption, it is reasonable to expect that most future induction distributions $\tilde{\mathbb{P}}$ can be represented as combinations of the basis distributions $\mathbb{P}_g$ with $g \in \mathcal{G}$. As shown in (Sagawa et al., 2020), evaluating the worst-case reward over all $m$ groups in $\mathcal{G}$ is equivalent to evaluating the reward for the worst-case distribution within the ambiguity set $\mathfrak{M}$ defined in (5).

**Lemma 3.1.** *Consider an ambiguity set $\mathfrak{M}$ formed by $\mathbb{P}_g$s as defined in (5). For any state-action pair $(s, a) \in \mathcal{S} \times \mathcal{A}$, the worst-case expected reward satisfies:*

$$\inf_{g \in \mathcal{G}} \mathbb{E}_{X \sim \mathbb{P}_g} [r(s, a; X)] = \inf_{\mathbb{P} \in \mathfrak{M}} \mathbb{E}_{X \sim \mathbb{P}} [r(s, a; X)], \tag{6}$$

*where $\mathcal{G}$ denotes the set of group indices.*

The proof of Lemma 3.1 is provided in Appendix A. The above lemma highlights a key advantage of group DRO: while general DRO problems are infinite-dimensional and computationally challenging, group DRO reduces the optimization to a finite-dimensional problem over $m$ groups. This reduction makes training of distributionally robust MARL (DRMARL) computationally tractable.

## 3.2. DRMARL with Group DRO

In the MARL framework described in Section 2, the policy parameters $\theta$ are optimized as follows:

$$\theta^* := \arg\min_{\theta \in \Theta} \mathbb{E}_{X \sim \mathbb{P}} [L(\theta; X)]. \tag{7}$$

Directly applying group DRO to MARL then leads to:

$$\tilde{\theta} := \arg\min_{\theta \in \Theta} \left\{ \max_{g \in \mathcal{G}} \mathbb{E}_{X \sim \mathbb{P}_g} [L(\theta; X)] \right\}. \tag{8}$$

However, conventional group DRO approaches are not directly applicable to MARL problems, as minimizing the worst-case Bellman error across groups does not necessarily yield a policy that is optimal under worst-case rewards. This disconnect arises because the worst-case Bellman error captures the maximum deviation from the target Q-function across groups, but it does not guarantee convergence to a robust optimal Q-function. To address this limitation, we introduce the distributionally robust Bellman operator, formally defined in the following result.

**Lemma 3.2.** *For an ambiguity set $\mathfrak{M}$ defined in (5) with group set $\mathcal{G}$, the distributionally robust (DR) Bellman operator is given by:*

$$\tilde{\mathcal{T}}_{\mathcal{G}}(\tilde{Q})(s, a) = \inf_{g \in \mathcal{G}} \mathbb{E}_{X \sim \mathbb{P}_g} [r(s, a; X)] + \gamma \max_{a'} \tilde{Q}(s', a'),$$

*where $\tilde{Q}$ is the distributionally robust Q-function. Moreover, the DR Bellman operator $\tilde{\mathcal{T}}_{\mathcal{G}}$ is also a contraction mapping.*

The proof of Lemma 3.2 is provided in Appendix A. Accordingly, the distributionally robust loss is given by

$$\begin{aligned} \tilde{L}(\theta; X) := \mathbb{E}_{s,a,r,a'} \Big[ \big( \tilde{Q}(s, a; \theta) \\ - \inf_{g \in \mathcal{G}} \mathbb{E}_{X \sim \mathbb{P}_g} [r(s, a; X)] - \gamma \max_{a'} \bar{\tilde{Q}}(s', a'; \bar{\theta}) \big)^2 \Big], \end{aligned} \tag{9}$$

and the distributionally robust parameters $\tilde{\theta}_{\mathcal{G}}$ are obtained by solving

$$\tilde{\theta}_{\mathcal{G}} := \arg\min_{\theta \in \Theta} \tilde{L}(\theta; X). \tag{10}$$

# 4. Contextual Bandit-based Worst-Case Reward Estimator for DRMARL

Solving the MARL group DRO problem (10) is theoretically feasible since the worst-case reward can be evaluated for each $(s, a)$ by exhaustively searching over all distribution groups $\mathcal{G}$. However, this approach is inefficient when the number of groups is large and forward simulation in the environment is costly. This is particularly the case in the dynamic chute mapping problem, where millions of packages are sorted across many destinations. Common group DRO techniques, such as soft reweighting (Sagawa et al.,

---

**Algorithm 1** CB-based Worst-Case Reward Estimator

---

1: **Input:** Learning rate $l_{\text{CB}}$, initial parameters $\psi_0$, induction distribution groups $\mathcal{G}$, MARL policy with $Q_{\text{MARL}}$, exploration rate $\varepsilon_{\text{CB}}$
2: **Initialize:** $\psi \leftarrow \psi_0$, replay buffer $\mathcal{D}_{\text{CB}} \leftarrow \emptyset$
3: **for** episode $= 1, ..., k_{\text{CB}}$ **do**
4:     Initialize the environment with random group $g' \in \mathcal{G}$ and observe initial state $s_0$
5:     **for** time step $t = 0, ..., T$ **do**
6:         Select action $a_t \leftarrow \arg\max_{a \in \mathcal{A}} Q_{\text{MARL}}(s_t, a)$
7:         With probability $\varepsilon_{\text{CB}}$, select $g' \sim \text{Uniform}(\mathcal{G})$; otherwise, $g' \leftarrow \arg\min_{g \in \mathcal{G}} Q_{\text{CB}}(s_t, a_t, g; \psi)$
8:         Execute action $a_t$, observe reward $r_t$ and next state $s_{t+1}$ using group $g'$
9:         Store transition $(s_t, a_t, g', r_t, s_{t+1})$ in $\mathcal{D}_{\text{CB}}$
10:        Sample mini-batch from $\mathcal{D}_{\text{CB}}$ and update $\psi$:
11:        $\psi \leftarrow \psi - l_{\text{CB}} \nabla_\psi L_{\text{CB}}(\psi)$
12:        $s_t \leftarrow s_{t+1}$, reduce $\varepsilon_{\text{CB}}$
13:     **end for**
14: **end for**
15: **Output:** Optimized CB estimator parameters $\psi^*$

---

**Algorithm 2** DRMARL with CB-based Worst-Case Reward Estimator

---

1: **Input:** Learning rate $l_r$, initial parameters $\theta_0$, induction distribution groups $\mathcal{G}$, pre-trained CB-based estimator $Q_{\text{CB}}$, exploration rate $\varepsilon$
2: **Initialize:** $\theta \leftarrow \theta_0$, replay buffer $\mathcal{D} \leftarrow \emptyset$
3: **for** episode $= 1, ..., k$ **do**
4:     Initialize the environment with random group $g' \in \mathcal{G}$ and observe initial state $s_0$
5:     **for** time step $t = 0, ..., T$ **do**
6:         With probability $\varepsilon$, select $a_t \sim \text{Uniform}(\mathcal{A})$; otherwise, $a_t \leftarrow \arg\max_{a \in \mathcal{A}} \tilde{Q}(s_t, a; \theta)$
7:         Estimate worst-case distribution group using CB: $g' \leftarrow \arg\min_{g \in \mathcal{G}} Q_{\text{CB}}(s_t, a_t, g)$
8:         Execute $a_t$, observe reward $r_t$ and next state $s_{t+1}$ using group $g'$
9:         Store transition $(s_t, a_t, g', r_t, s_{t+1})$ in $\mathcal{D}$
10:        Sample mini-batch from $\mathcal{D}$ and update parameters: $\theta \leftarrow \theta - l_r \nabla_\theta \tilde{L}(\theta)$
11:        $s_t \leftarrow s_{t+1}$, reduce $\varepsilon$
12:     **end for**
13: **end for**
14: **Output:** Optimized DRMARL policy parameters $\tilde{\theta}_{\mathcal{G}}$

---

2020), may not perform well in MARL because, unlike regression tasks, the data distribution depends on the agent's policy. As the policy evolves, the groups that are underrepresented or perform poorly will change dynamically. This dynamic nature of the problem makes it challenging to apply static or even adaptive reweighting schemes, which assume a relatively stable data distribution.

To improve training efficiency, we propose *a novel contextual bandit-based worst-case reward distribution estimator* that learns to identify the worst-case distribution group $g \in \mathcal{G}$ for each state-action pair $(s, a)$ by training a contextual bandit (CB) model (Li et al., 2010; Russo et al., 2018).

### 4.1. CB-based Worst-Case Reward Estimator

The CB treats the current state-action pair $(s, a)$ as context, and its arms corresponding to the distribution groups in the set $\mathcal{G}$. The goal is to identify the group $g$ that minimizes the reward $\mathbb{E}_{X \sim \mathbb{P}_g}[r(s, a; X)]$, which represents the worst-case reward among all groups. The CB is constructed as follows:

**Context space:** $\mathcal{S} \times \mathcal{A}$, where $(s, a) \in \mathcal{S} \times \mathcal{A}$ represents the state and the chute mapping actions at each step.

**Action space:** $\mathcal{G} = \{1, 2, \ldots, m\}$, where $g \in \mathcal{G}$ denotes a group associated with an induction distribution.

**Reward:** A reward function $r : (\mathcal{S} \times \mathcal{A}) \times \mathcal{G} \to \mathbb{R}$, where $r(s, a; X)$ represents the observed reward for choosing distribution $\mathbb{P}_g$ at the current state-action pair $(sa)$.

The CB is represented by a Q-function that approximates the expected reward of choosing distribution $\mathbb{P}_g$ given a context $(s, a)$. For this purpose, we use an independent DQN (Mnih et al., 2015):

$$Q_{\text{CB}}(s, a, g; \psi) = \mathbb{E}_{X \sim \mathbb{P}_g}[r(s, a; X)], \qquad (11)$$

where the reward function $r(s, a; X)$ is observed after running a single-step forward simulation with $(s, a)$ under the distribution $\mathbb{P}_g$. The $Q_{\text{CB}}$ function is learned by minimizing the following loss (see the detailed training process in Algorithm 1):

$$
\begin{aligned}
L_{\text{CB}}(\psi) := \\
\mathbb{E}_{s,a}\left[\left(Q_{\text{CB}}(s, a, g; \psi) - \mathbb{E}_{X \sim \mathbb{P}_g}[r(s, a; X)]\right)^2\right].
\end{aligned} \qquad (12)
$$

In Algorithm 1, the exploration of state-action pairs $(s, a)$ is guided by the existing MARL policy using $Q_{\text{MARL}}$, which ensures sufficient exploration of the context space for the chute-mapping problem. For other applications, different exploration policies can be employed, such as random action selection, to ensure adequate coverage of the state-action space.

### 4.2. DRMARL with CB-based Worst-Case Reward Estimator

Once the $Q_{\text{CB}}$ function has been learned, we can rewrite the

Table 1: Comparison of key sortation metrics across policies, averaged over $m = 9$ groups, each evaluated over 100 runs. See Appendix D.1 for detailed results.

| Policy | Recirculation Rate ($\downarrow$) | Throughput ($\uparrow$) | Recirculation Amount ($\downarrow$) |
|---|---|---|---|
| MARL | $2.16\% \pm 2.35\%$ | 11740.98 | 259.02 |
| DRMARL (random) | $1.56\% \pm 1.45\%$ | 11812.77 | 187.23 |
| DRMARL (with $Q_{CB}$) | $\mathbf{0.56\% \pm 0.18\%}$ | **11932.21** | **67.79** |
| DRMARL (exhaustive) | $0.55\% \pm 0.13\%$ | 11933.68 | 66.32 |
| MARL (group-specific) | $0.53\% \pm 0.14\%$ | 11936.60 | 63.40 |

distributionally robust loss (9) as:

$$
\tilde{L}(\theta; X) = \mathbb{E}_{s,a,r,a'} \Big[ \big( \tilde{Q}(s,a;\theta) \\
- \mathbb{E}_{X \sim \mathbb{P}_{g'}} \left[ r(s,a;X) \right] - \gamma \max_{a'} \bar{\tilde{Q}}(s',a';\bar{\theta}) \big)^2 \Big], \quad (13)
$$

where $g' = \arg\min_{g \in \mathcal{G}} Q_{CB}(s,a,g)$ is the index of the distribution group with the estimated worst-case reward.

The training procedure of DRMARL is shown in Algorithm 2. The key difference compared to MARL training is that DRMARL aims to train a DQN that estimates the worst-case return across all groups, while MARL aims to estimate the observed return only for a specific induction distribution. Moreover, in contrast to traditional group DRO, the index of the worst-case distribution $g'$ in DRMARL is not obtained via exhaustive search; instead, it is estimated by $Q_{CB}$ given the state-action pair $(s, a)$. The independent Q-network $Q_{CB}$ is learned beforehand using Algorithm 1 and remains unchanged during DRMARL training. While it may seem counterintuitive that $Q_{CB}$ estimates the worst-case group after $\tilde{Q}$ selects an action, this design enables $Q_{CB}$ to provide the worst-case expected return for each $(s, a)$ pair, thereby enabling the learning of a robust policy.

## 5. Numerical Experiments

In this section, we demonstrate the effectiveness of the proposed DRMARL policy under OOD induction changes in both a simplified simulation and a large-scale Amazon robotic sortation warehouse simulation environment.

### 5.1. Simplified Robotic Sortation Warehouse Simulation

In the simplified simulation environment, there are 10 eject chutes, one recirculation chute, and 20 unique destinations. Packages arrive at the sortation warehouse according to the induction data $X$ generated from an induction distribution $\mathbb{P}$. When packages exceed the capacities of eject chutes, they are sent to the recirculation chute. One training or testing episode consists of 5 hours, with each time step being 30 minutes long, after which the environment is reset. An eject chute can be reallocated at each time step. The implementation details are provided in Appendix B.

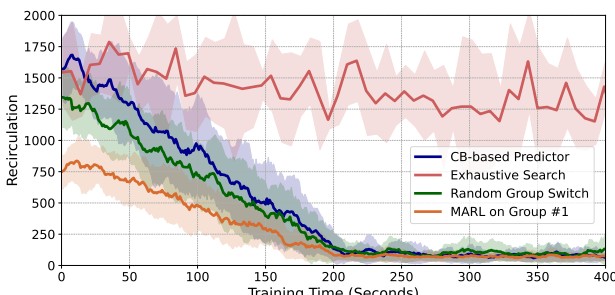

Figure 5: Training efficiency comparison in simplified warehouse.

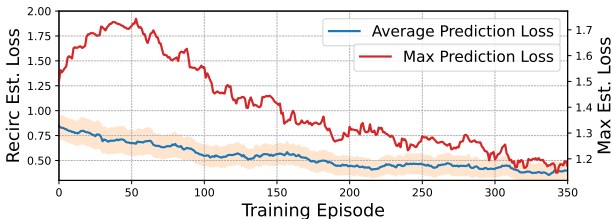

Figure 6: Training recirculation rate estimation loss (12) for CB in simplified robotic sortation warehouse.

We train the DRMARL policy over 300 episodes using training data generated from 9 distinct induction distribution groups. Similarly, the regular MARL policies are trained for 300 episodes, each on one of the same groups. Due to the stochastic nature of the induction-generating distributions, the induction data varies across simulation trials.

Key metric comparisons are shown in Table 1. DRMARL with $Q_{CB}$ achieves the best performance across all metrics. For reference, the last row shows the theoretical upper bound achieved by group-specific MARL policies, trained and evaluated on the same group. DRMARL performs only marginally below this oracle baseline, highlighting its strong balance between individual performance and robustness. Compared to random group selection and exhaustive worst-case search, $Q_{CB}$ efficiently identifies challenging groups during training and achieves comparable robustness with significantly lower computational cost. These results demonstrate that $Q_{CB}$ enables DRMARL to focus on worst-case distributions effectively, yielding robust policies without sacrificing efficiency. We refer to Appendix D.1 for a detailed discussion of these results.

Figure 6 shows the training progress of $Q_{CB}$, with clear reductions in both average and maximum recirculation rate estimation errors over time. Training efficiency comparisons across methods are shown in Figure 5. DRMARL with $Q_{CB}$ converges in under 300 seconds, while exhaustive worst-case search exceeds 2900 seconds due to its full traversal of all groups. In contrast, $Q_{CB}$ achieves comparable robustness at significantly lower computational cost,

Table 2: Relative improvement (↑) of DRMARL over MARL baseline, averaged across $m = 21$ groups.

| Policy | Recirc Rate Reduction | Throughput Increase | Recirc Amount Reduction |
|---|---|---|---|
| DRMARL | 79.97% | 5.62% | 33.64% |
| MARL (group-specific) | 85.42% | 9.80% | 40.50% |

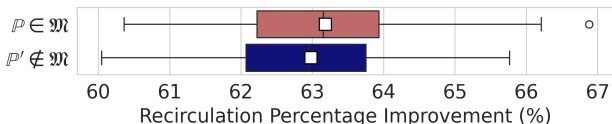

Figure 7: Recirculation rate improvement of DRMARL over two equally-performing MARL policies trained on distributions inside and outside $\mathfrak{M}$.

Table 3: Relative reduction (↑) in CVaR, worst-case recirculation amount, and average recirculation rate across different Robust RL approaches.

| Method | CVaR Reduction | Worst-case Recirc Reduction | Recirc Rate Reduction |
|---|---|---|---|
| Adversarial Agent | 61.11% | 53.57% | 66.19% |
| Adversarial Environment | 58.48% | 23.68% | 77.09% |
| Robustified Reward | 58.97% | 60.23% | 70.43% |
| DRMARL | **77**.09% | **76**.61% | **79**.97% |

validating its ability to efficiently identify worst-case groups. Although DRMARL with random group selection converges faster, it lacks the worst-case guarantees provided by $Q_{\text{CB}}$. Group-specific MARL converges fastest overall, reflecting the simplicity of its single-group training setup.

## 5.2. Large-Scale Robotic Sortation Warehouse Simulation

### 5.2.1. IMPLEMENTATION DETAILS

In the large-scale simulation environment, there are 187 eject chutes, one recirculation chute, and 120 unique destinations. Packages arrive at the sortation warehouse according to the corresponding induction data $X$ generated from the induction distribution $\mathbb{P}^2$. When packages exceed chute capacities or miss departure transportation schedules, they are sent to the recirculation chute and added to the queue of new packages at the next time step. One training/testing episode lasts 11 hours, with each time step lasting five minutes, after which the environment is reset. Every five minutes, destinations are assigned to chutes that become available for reallocation

We train the DRMARL policy over 200 episodes using training data generated from 21 distinct induction distribution groups spanning several years. Similarly, the regular MARL policy is trained for 200 episodes using induction data from Year 4. For testing, we evaluate both policies on newly generated induction data from 21 distinct distribution groups, conducting five experiments per group. Due to the stochastic nature of the induction-generating distributions, the test induction data remains unseen during training for both policies.

### 5.2.2. ROBUSTNESS OF THE CHUTE MAPPING POLICY

Table 2 presents the average relative performance improvement of DRMARL across all 21 distribution groups, using MARL as the baseline. DRMARL demonstrates robust performance across all induction groups, consistently outperforming the baseline MARL policy. For reference,

---

[2] Due to common industry confidentiality practices, we cannot disclose the specific data source and report only relative performance improvements. The data represents realistic package flow patterns typical of Amazon robotic sortation facilities.

the bottom row shows the theoretical optimal performance achieved by training and testing group-specific MARL policies on each individual group. As expected, DRMARL performs *marginally* below these group-specific MARL policies, illustrating the trade-off between robustness and individual group performance. Detailed results are provided in Appendix D.2.

As more production data becomes available, the distribution groups can be expanded to better capture year-to-year shifts without increasing the training complexity of DRMARL. To assess robustness beyond the ambiguity set $\mathfrak{M}$, we evaluate the learned policy on induction distributions $\mathbb{P}'$ where $\mathbb{P}' \notin \mathfrak{M}$. As shown in Figure 7, the robust policy generalizes well to these out-of-distribution settings. Notably, the Type-1 Wasserstein distance between $\mathbb{P}'$ and the closest distribution in $\mathfrak{M}$ is 818.19, compared to an average intra-group distance of 542.96 within $\mathfrak{M}$.

We also compare DRMARL against three robust RL baselines, with results summarized in Table 3: (i) an adversarial agent that deliberately blocks available chutes (Mandlekar et al., 2017); (ii) an adversarial environment with perturbed transition dynamics (Pinto et al., 2017); and (iii) a robust MDP formulation that uses a worst-case reward function over an uncertainty set (Wiesemann et al., 2013). We evaluate performance using three metrics: cumulative recirculation over the 11-hour simulation; worst-case instantaneous recirculation; and Conditional Value-at-Risk (CVaR) at the 5% level (Rockafellar & Uryasev, 2000). To compute CVaR, we collect instantaneous recirculation values at all time steps and estimate their empirical distribution. CVaR at 5% corresponds to the expected recirculation conditional on exceeding the 95th percentile, capturing the expected severity of the worst-case recirculation (higher values are more undesirable). While some robust RL approaches achieve

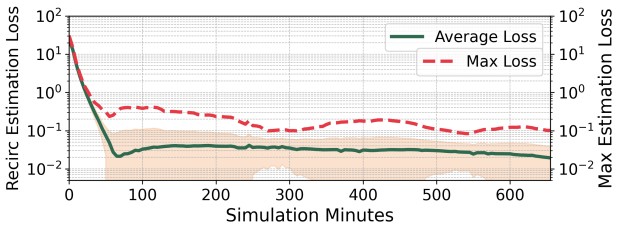

Figure 8: $Q_{\text{CB}}$ estimation loss for recirculation percentage, averaged over 25 test simulations.

cumulative recirculation comparable to DRMARL, only DRMARL consistently improves the worst-case recirculation performance by explicitly optimizing for it. DRMARL outperforms all robust RL baselines due to its distributional awareness, which yields more accurate worst-case reward estimation and robust action-value functions. This robustness is critical in practice, as sudden surges in recirculation can lead to robot congestion and delays in the sorting process. Since robot congestion is not explicitly modeled in the simulator, the real-world advantage of DRMARL over standard robust RL methods is likely even greater.

### 5.2.3. CB-BASED WORST-CASE REWARD ESTIMATOR

Following Section 4, we train an independent Q-network, $Q_{\text{CB}}$, to estimate the worst-case recirculation (reward) for all state-action pairs $(s, a)$ across groups $\mathcal{G}$. The trained $Q_{\text{CB}}$ achieves high accuracy, with estimation errors below $1\%$ of the recirculation rate, enabling reliable identification of worst-case scenarios among the groups $g \in \mathcal{G}$.

During each day's 11-hour simulation, as illustrated in Figure 8, the estimation accuracy improves substantially after the first hour. Although initially suboptimal, $Q_{\text{CB}}$'s performance remains sufficient for DRMARL training, since the impact of the worst-case distributions on recirculation becomes more pronounced in later stages when fewer chutes are available.

### 5.2.4. EFFICIENT TRAINING WITH CB-BASED WORST-CASE REWARD ESTIMATOR

The CB-based worst-case reward estimator, $Q_{\text{CB}}$, substantially improves training efficiency by eliminating the need for exhaustive group evaluation at each time step, reducing the computational complexity of the worst-case group identification from $\mathcal{O}(m)$ to $\mathcal{O}(1)$. As demonstrated in Figure 9, training with $Q_{\text{CB}}$ achieves significantly faster convergence compared to exhaustive evaluation over $\mathcal{G}$, which requires approximately $924$ hours on a cloud instance with 64 vCPUs (Intel Xeon Scalable 4th generation) and 128 GB RAM. The lightweight Q-network updates enabled CPU-only training, with most of the computation time spent on environment simulation. This efficiency advantage becomes even more

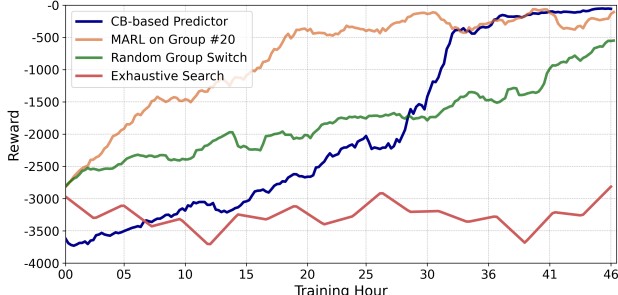

Figure 9: Training computational efficiency comparison in large-scale Amazon robotic sortation warehouse simulation environments.

pronounced when dealing with complex environments or larger group sets.

Figure 9 also compares the training efficiency of different approaches. The group-specific MARL, which trains on group #20, shows the fastest convergence due to its simplified learning objective. Among distributionally robust approaches, DRMARL with random group selection ($g' \leftarrow \text{random}(\mathcal{G})$) initially converges faster than other robust variants but achieves suboptimal robustness, since it may miss critical worst-case scenarios. DRMARL with $Q_{\text{CB}}$ strikes a balance between training speed and performance, converging significantly faster than exhaustive search while maintaining near-optimal worst-case performance guarantees. As expected, DRMARL with exhaustive search over all distribution groups requires the longest training time, though it serves as a valuable baseline for validating the efficiency of our $Q_{\text{CB}}$-based approach.

## 6. Conclusion

In this paper, we introduced DRMARL, a framework that integrates group DRO into MARL to enhance policy robustness against OOD distribution shifts in warehouse sortation systems. To address the computational cost of identifying the worst-case group, we developed a CB-based estimator that significantly reduces the complexity of worst-case identification from $\mathcal{O}(m)$ to $\mathcal{O}(1)$. Experimental results from both simplified and large-scale warehouse environments demonstrate that DRMARL achieves robust and near-optimal performance across all distribution groups while maintaining computational efficiency. The framework also shows strong generalization to distributions outside the training set, and its design principles can be extended to other MARL applications where distributional robustness is crucial.

# Acknowledgments

We would like to express our sincere gratitude to our colleagues at Amazon Robotics for their support and valuable contributions throughout this research. In particular, we extend special thanks to Rahul Chandan and Mouhacine Benosman for their insightful feedback and constructive discussions related to this work.

# Impact Statement

This work advances both machine learning research and sustainable warehouse operations. The proposed DRMARL framework reduces package recirculation, directly contributing to lower energy consumption and carbon emissions in e-commerce logistics. From a societal perspective, more efficient package sorting translates to both lower consumer costs and faster delivery times, particularly benefiting those who rely on e-commerce for essential goods. For warehouse operations, the system creates more predictable workflows, potentially reducing physical strain on workers and improving labor conditions. The framework's robust performance across varying distribution patterns ensures consistent service levels regardless of package volume or ordering patterns, promoting equitable service delivery across different communities. Beyond warehouses, our approach to distributional robustness contributes to the broader development of reliable AI systems for real-world applications where performance consistency is crucial.

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

# Appendix

# A. Proofs of Theoretical Results

### A.1. Proof of Lemma 3.1

Recall the definition of $\mathfrak{M}$ in (5) and for any probability distribution $\mathbb{P} \in \mathfrak{M}$, we have

$$
\begin{aligned}
\mathbb{E}_{X\sim\mathbb{P}}\left[r(s,a;X)\right] &= \int r(s,a;X)\,\mathbb{P}(X)\,dX \\
&= \int r(s,a;X)\left(\sum_{g=1}^{m} q_g\,\mathbb{P}_i(X)\right)dX \\
&= \sum_{g=1}^{m} q_g \int r(s,a;X)\,\mathbb{P}_i(X)\,dX = \sum_{g=1}^{m} q_g\,\mathbb{E}_{X\sim\mathbb{P}_g}\left[r(s,a;X)\right].
\end{aligned}
\tag{14}
$$

Then, taking the infimum on both sides yields:

$$
\begin{aligned}
\inf_{\mathbb{P}\in\mathfrak{M}}\mathbb{E}_{X\sim\mathbb{P}}\left[r(s,a;X)\right] &= \inf_{q\in\Delta_m}\sum_{g=1}^{m} q_g\mathbb{E}_{X\sim\mathbb{P}_g}\left[r(s,a;X)\right] \\
&= \inf_{g\in\mathcal{G}}\mathbb{E}_{X\sim\mathbb{P}_g}\left[r(s,a;X)\right],
\end{aligned}
\tag{15}
$$

since optimum of a linear program over simplex $\Delta_m$ is obtained at vertices. ∎

### A.2. Proof of Lemma 3.2

In order to find the group distributionally robust action-value function $\tilde{Q}$, we consider the worst-case immediate reward at each state-action pair $(s,a)$ as:

$$
\tilde{r}(s,a) = \min_{g\in\mathcal{G}}\mathbb{E}_{X\sim\mathbb{P}_g}\left[r(s,a;X)\right] \leq \mathbb{E}_{X\sim\mathbb{P}}\left[r(s,a;X)\right],
\tag{16}
$$

for all unknown distribution $\mathbb{P}\in\mathfrak{M}$. Then, the worst-case return is given by:

$$
\tilde{R}_t = \sum_{k=0}^{\infty}\gamma^k\,\tilde{r}(s,a) \leq \sum_{k=0}^{\infty}\gamma^k\,\mathbb{E}_{X\sim\mathbb{P}_k}[r_{t+k+1}(s,a;X)],
\tag{17}
$$

where $\gamma\in[0,1]$ is the discount factor and the inequality holds for all possible sequences $\{\mathbb{P}_k\}_{k=0}^{\infty}$ with $\mathbb{P}_k\in\mathfrak{M}$. Then, the worst-case action-value function under policy $\pi$ can be expressed as:

$$
\begin{aligned}
\tilde{Q}^\pi(s,a) &= \mathbb{E}_\pi\left[\tilde{r}(s,a) + \gamma\tilde{Q}^\pi(s_{t+1},a_{t+1})\mid s_t=s,a_t=a\right] \\
&= \mathbb{E}_{s'\sim P(\cdot|s,a)}\left[\tilde{r}(s,a) + \gamma\mathbb{E}_{a'\sim\pi(\cdot|s')}\tilde{Q}^\pi(s',a')\right],
\end{aligned}
\tag{18}
$$

and the optimal worst-case action-value function satisfies the Bellman optimality equation:

$$
\tilde{Q}^*(s,a) = \tilde{r}(s,a) + \gamma\,\mathbb{E}_{s'\sim P(\cdot|s,a)}\left[\max_{a'}\tilde{Q}^*(s',a')\right].
\tag{19}
$$

Following the result from (Liu et al., 2022), the distributionally robust Bellman Operator with ambiguity sets $\mathfrak{P}$ and $\mathfrak{R}$ is given by

$$
\tilde{\mathcal{T}}_{\mathfrak{P},\mathfrak{R}}(\tilde{Q})(s,a) = \inf_{\substack{p_{s,a}\in\mathfrak{P}\\r_{s,a}\in\mathfrak{R}}}\left\{\mathbb{E}_{r_{s,a}}[r(s,a)] + \gamma\,\mathbb{E}_{p_{s,a}}\left[\max_{a'}\tilde{Q}(s',a')\right]\right\},
\tag{20}
$$

where $p_{s,a}$ and $r_{s,a}$ denote the distributions of state transitions probabilities and reward functions respectively, with their corresponding ambiguity sets $\mathfrak{P}$ and $\mathfrak{R}$. Since the distribution shift in the random variable $X$ only affects the reward, i.e.,

the distribution of the reward function $r_{s,a}$, we have:

$$
\begin{aligned}
\tilde{\mathcal{T}}_{\mathfrak{R}}(\tilde{Q})(s,a) &= \inf_{r_{s,a} \in \mathfrak{R}} \left\{ \mathbb{E}_{r_{s,a}}[r(s,a)] + \gamma \left[ \max_{a'} \tilde{Q}(s',a') \right] \right\} \\
&= \tilde{r}(s,a) + \gamma \max_{a'} \tilde{Q}(s',a') \\
&= \inf_{g \in \mathcal{G}} \left\{ \mathbb{E}_{X \sim \mathbb{P}_g}[r(s,a;X)] \right\} + \gamma \max_{a'} \tilde{Q}(s',a') = \tilde{\mathcal{T}}_{\mathcal{G}}(\tilde{Q})(s,a).
\end{aligned}
\tag{21}
$$

We further show the distributionally robust Bellman operator is a contraction mapping under the $\ell_\infty$ norm. Consider two arbitrary robust action-value functions $\tilde{Q}_1$ and $\tilde{Q}_2$ such that

$$
\begin{aligned}
\tilde{\mathcal{T}}_{\mathcal{G}}(\tilde{Q}_1)(s,a) &= \tilde{r}(s,a) + \gamma \max_{a'} \tilde{Q}_1(s',a') \\
\tilde{\mathcal{T}}_{\mathcal{G}}(\tilde{Q}_2)(s,a) &= \tilde{r}(s,a) + \gamma \max_{a'} \tilde{Q}_2(s',a').
\end{aligned}
\tag{22}
$$

Finding the difference yields

$$
|\tilde{\mathcal{T}}_{\mathcal{G}}(\tilde{Q}_1)(s,a) - \tilde{\mathcal{T}}_{\mathcal{G}}(\tilde{Q}_2)(s,a)| \leq \gamma \max_{s',a'} |\tilde{Q}_1(s',a') - \tilde{Q}_2(s',a')|,
\tag{23}
$$

and taking the maximum over all feasible state-action pair $(s,a)$ implies

$$
\|\tilde{\mathcal{T}}_{\mathcal{G}}(\tilde{Q}_1) - \tilde{\mathcal{T}}_{\mathcal{G}}(\tilde{Q}_2)\|_{\ell_\infty} \leq \gamma \|\tilde{Q}_1 - \tilde{Q}_2\|_{\ell_\infty}.
\tag{24}
$$

Since $\gamma \in [0,1]$, this establishes that DR Bellman operator is a contraction mapping under the $\ell_\infty$ norm. By Banach's Fixed Point Theorem (Rudin, 2021), there exists a unique fixed point $\tilde{Q}^*$ such that $\tilde{\mathcal{T}}_{\mathcal{G}}(\tilde{Q}^*) = \tilde{Q}^*$. Consequently, iteratively applying the operator ensures convergence to $\tilde{Q}^*$, proving the stability of the robust Q-learning algorithm. This contraction property of the DR Bellman operator was also addressed in (Iyengar, 2005) when the ambiguity set is defined for transition probability. ∎

## B. Implementation Details for Simplified Robotic Sortation Warehouse

We define a Markov game for $N$ agents (representing unique destinations) by the tuple $\left(N, \mathcal{S}, \{\mathcal{O}^i\}_{i=1}^N, \{\mathcal{A}^i\}_{i=1}^N, P, \{r^i\}_{i=1}^N, \gamma, \rho_0\right)$, where:

(a) **Agents:** The set of $N$ agents, each corresponding to a unique destination.

(b) **State Space:** $\mathcal{S}$ denotes the joint state space.

(c) **Observation Space:** For each agent $i$, $\mathcal{O}^i \subset \mathcal{S}$ represents its local observation at each time step, consisting of:

- The total number of assignable chutes (uniform across all agents)
- The number of chutes currently assigned to agent $i$

(d) **Action Space:** For each agent $i$, $\mathcal{A}^i \subset [0,1]$ represents its action space, where each action determines if a new chute will be allocated. An action value of 1 indicates the assignment of a new chute to destination $i$. The joint action space is defined as $\mathcal{A} = \prod_{i=1}^N \mathcal{A}^i$.

(e) **Transition Probability:** $P : \mathcal{S} \times \mathcal{A} \times \mathcal{S} \to [0,1]$ specifies the probability of transitioning between states, representing the likelihood of packages being either successfully sorted or diverted to the recirculation buffer.

(f) **Reward Function:** For each agent $i$, $r^i : \mathcal{S} \times \mathcal{A} \times \mathcal{X} \to \mathbb{R}$ defines the reward function, which penalizes the number of packages in the recirculation buffer resulting from the current chute allocation.

The model is completed with discount factor $\gamma \in (0,1)$ and initial state distribution $\rho_0$. In Section 5.1, we employ the Value Decomposition Network (VDN) (Shen et al., 2023) combined with budget constraints in computing joint actions.

In the simplified robotic sortation environment, we fix the total induction volume at each time step to 1200 packages. The number of incoming packages for each destination $i = 1, \ldots, N$ follows an unknown normal distribution $\mathcal{N}(\mu, \sigma)$. For each destination $i$, the probability that an incoming package is assigned to destination $i$ is given by:

$$\mathbb{P}\Big\{\text{incoming package belongs to } i\Big\} = \frac{\Phi(\frac{i-\mu}{\sigma}) - \Phi(\frac{i-1-\mu}{\sigma})}{\Phi(\frac{N-\mu}{\sigma}) - \Phi(\frac{-\mu}{\sigma})}, \tag{25}$$

where $\Phi(z) = \frac{1}{\sqrt{2\pi}} \int_{-\infty}^{z} e^{-\frac{t^2}{2}} \, dt$ is the cumulative distribution function of the standard normal distribution. This formulation ensures $\sum_{i=1}^{N} \mathbb{P}\{\text{incoming package belongs to } i\} = 1$. The distribution of packages across destinations is then determined by sampling 1200 packages according to the probabilities defined in (25) at each time step.

In Section 5.1, we assume the destination transportation vehicle has infinite capacity, meaning packages enter the recirculation buffer only when incoming packages are destined for a location without an assigned eject chute. In this example, we construct the ambiguity set as:

$$\mathfrak{M} := \Big\{ \tilde{\mathbb{P}} = \sum_{g=1}^{m} q_g \, \mathbb{P}_g \mid q \in \Delta_m \Big\}, \tag{26}$$

where each $\mathbb{P}_g$ represents a normal distribution $\mathcal{N}(\mu_g, \sigma)$ with different $\mu_g$s. For our simulation, we construct the ambiguity set using $m = 9$ groups with means $\mu_g \in \{-4, -3, \ldots, 0, \ldots, 4\}$, standard deviation $\sigma = 2$, and index set $\mathcal{G} = \{1, 2, \ldots, 9\}$.

## C. Implementation Details for Large-Scale Robotic Sortation Warehouse

### C.1. MARL Structure

We define a Markov game over $N$ agents (unique destinations) by the tuple $\big(N, \mathcal{S}, \{\mathcal{O}^i\}_{i=1}^{N}, \{\mathcal{A}^i\}_{i=1}^{N}, P, \{r^i\}_{i=1}^{N}, \gamma, \rho_0\big)$, where:

(a) **Agents:** $N$ agents, each corresponding to a unique destination.

(b) **State Space:** $\mathcal{S}$ denotes the joint state space.

(c) **Observation Space:** For each agent $i$, $\mathcal{O}^i \subset \mathcal{S}$ represents its local observation at each time step, consisting of:
   - Number of packages recirculated until time $t$ for agent $i$
   - Total number of available chutes that can be assigned (uniform across all agents)
   - Number of chutes currently assigned to agent $i$

(d) **Action Space:** For each agent $i$, $\mathcal{A}^i$ represents its action space, determining the number of new chutes required. Actions take values in $[0, 10]$, where an action value $a$ indicates the assignment of $a$ new chutes to agent $i$ at that time step. The joint action space is defined as $\mathcal{A} = \prod_{i=1}^{N} \mathcal{A}^i$.

(e) **Transition Probability:** $P : \mathcal{S} \times \mathcal{A} \times \mathcal{S} \times \mathcal{X} \to [0, 1]$ specifies the probability of packages being either sorted by chutes or sent to the recirculation buffer. In the large-scale robotic sortation warehouse setting, the transition probability is a function of the induction distribution $\mathbb{P}$ with random variable $X \in \mathcal{X}$, which is addressed in detail in Appendix C.5.

(f) **Reward Function:** For each agent $i$, $r^i : \mathcal{S} \times \mathcal{A} \times \mathcal{X} \to \mathbb{R}$ defines the reward function, penalizing both the number of allocated chutes and the number of packages in the recirculation buffer. The recirculation is a function of the induction distribution, which is defined in Appendix C.2.

The model is completed with discount factor $\gamma \in (0, 1)$ and initial state distribution $\rho_0$.

### C.2. Induction Distribution

For a given sortation warehouse with $D$ destinations and $T$ time intervals (e.g., hours or minutes) in a day, we consider the random vector $X = \{X_1, \ldots, X_{DT}\} \in \mathbb{R}^{DT}$, where each $X_i \geq 0$ denotes the number of packages inducted for a specific

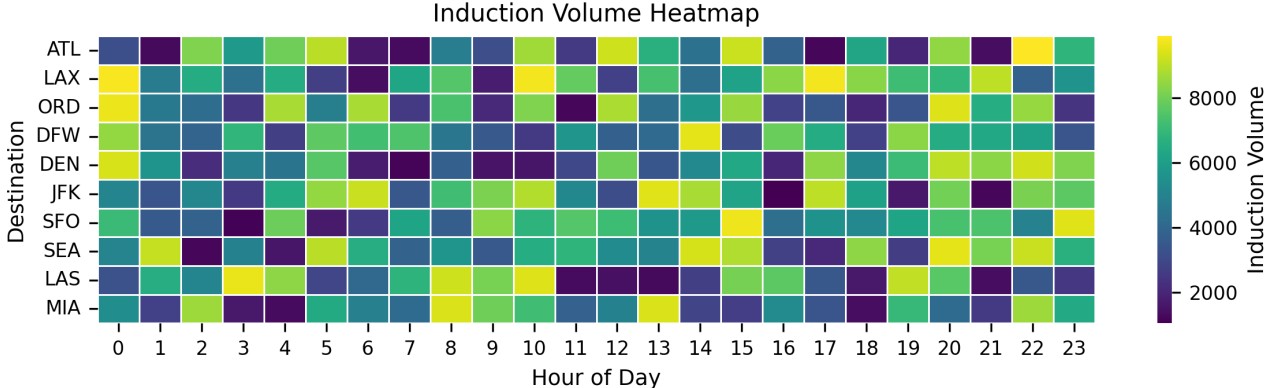

Figure 10: A realization of the random variable $X$. The values are synthetic and do not represent actual Amazon induction volumes.

destination-time pair $(d, t) \in \{1, \ldots, D\} \times \{1, \ldots, T\}$. That is, $X_{(d,t)}$ represents the number of packages inducted for destination $d$ during time interval $t$. A realization of the random variable $X$ is shown in Figure 10.

We model the daily induction pattern as a random variable $X$ drawn from an unknown distribution $\mathbb{P}$, which we refer to as the induction generating distribution. We assume that the total daily induction volume is fixed at $V$ across all days, as the MARL chute mapping policy is primarily influenced by the spatial-temporal distribution pattern rather than variations in total volume. The daily induction is thus generated by sampling a realization of $X$ from $\mathbb{P}$.

**Definition C.1** (Induction Generating Distribution). We define the induction generating distribution as a multinomial distribution

$$X = (X_1, \ldots, X_k) \sim \mathcal{M}(V, p_1, \ldots, p_k),$$

where $V \in \mathbb{N}$ is the total number of packages inducted in a day, and the probability vector $(p_1, \ldots, p_k)$ satisfies $p_i \geq 0$ and $\sum_{i=1}^k p_i = 1$. The support of $X$ is the set

$$\left\{ (z_1, \ldots, z_k) \in \mathbb{N}_0^k \;\middle|\; \sum_{i=1}^k z_i = V \right\}.$$

Its probability mass function is given by

$$\mathbb{P}(X_1 = z_1, \ldots, X_k = z_k) = \frac{n!}{z_1! z_2! \cdots z_k!} \, p_1^{z_1} \cdots p_k^{z_k},$$

for all $(z_1, \ldots, z_k)$ in the support.

For induction random variables $X$ corresponding to temporally proximate dates (e.g., within the same week), we assume they share a common induction generating distribution $\mathbb{P} = \mathcal{M}(V, p_1, \ldots, p_{DT})$. In practice, this distribution is estimated via the Sample Average Approximation (SAA) method (Kim et al., 2015) using historical induction data from the relevant dates. Specifically, we collect historical induction records from Years 1-4 and partition them into 21 groups based on week numbers. For each group $g$, we construct an empirical induction generating distribution $\mathbb{P}_g$ via SAA, modeled as a multinomial distribution using all induction observations within that group.

The group ambiguity set $\mathfrak{M}$ in (5) is then formed from the collection $\{\mathbb{P}_1, \mathbb{P}_2, \ldots, \mathbb{P}_{21}\}$. The probability $p_j$ of assigning an incoming package to the $j$-th destination-time pair is estimated from the corresponding empirical multinomial distribution $\mathbb{P}_g$. Given a total volume of $V$ packages, the daily induction is generated by a single sample from this distribution.

Amazon has publicly reported double-digit Year-over-Year (YoY) growth in retail sales in its Q4 earnings press releases over multiple consecutive years (Amazon.com, Inc., 2022; 2023; 2024). This sustained growth translates directly to the package sortation context, resulting in significant YoY increases in induction volumes. In practice, package volume and its spatial distribution exhibit substantial variability across both seasonal and annual timescales. Seasonal effects introduce

Table 4: Average Type-1 Wasserstein distances for each week compared to Week 1 of the same year.

| Year | Week 2 | Week 3 | Week 4 | Week 5 |
|---|---|---|---|---|
| Year 1 | 76.87 | 202.96 | 147.17 | 241.80 |
| Year 2 | 36.46 | 70.47 | 301.25 | 123.45 |
| Year 3 | 25.81 | 23.87 | 72.91 | 78.82 |
| Year 4 | 72.99 | 45.04 | 24.92 | 62.39 |

Table 5: Average Type-1 Wasserstein distance to Year 1 ($\downarrow$).

| Year 2 | Year 3 | Year 4 |
|---|---|---|
| 1172.85 | 585.98 | 288.86 |

predictable fluctuations driven by major sales events and holidays (e.g., Black Friday, Cyber Monday, and year-end gifting), while annual trends reflect more gradual changes due to evolving customer behavior, shifts in logistics strategies, and overall business expansion.

We further quantify these distributional changes using the empirical Type-1 Wasserstein distance (Villani et al., 2008), as shown in Table 4 and Table 5. Specifically, Table 4 compares four representative weeks to Week 1 within each year, revealing substantial seasonal variation in induction distributions. Additionally, we find that the Wasserstein distance between consecutive days is typically on the order of 20, indicating moderate short-term variability. Table 5 illustrates yearly distributional shifts by comparing the same temporal slices across different years to the baseline in Year 1. These results highlight that inter-annual changes tend to be even more pronounced than intra-annual seasonal effects, posing significant challenges for policies that rely on static assumptions or historical averages.

### C.3. Dimension Reduction of the State-Action Space

To manage the dimensionality of the state-action space, we decompose the joint Q-network into a sum of local Q-networks. Each local network captures the expected return of an individual agent's chute mapping actions, while the joint network represents the expected return of the complete chute assignment across all agents. Specifically, we express the joint Q-network as:

$$Q(s, a, \theta) = \sum_{i=1}^{N} Q'(i, s^i, a^i; \theta), \tag{27}$$

where the input space scales linearly with the number of agents. While this decomposition is similar to that proposed in (Shen et al., 2023), our approach learns a single shared $Q'$ network for all agents, rather than separate networks for each agent, resulting in improved computational efficiency.

### C.4. Feasibility of Joint Actions

In unconstrained settings, agents would simply select actions that maximize their individual Q-networks, with the joint action being the collection of these individual choices. However, the chute mapping problem introduces resource constraints, as agents must share a limited number of available chutes. This necessitates coordination to allocate resources optimally among the agents based on their state-action values.

Given a budget constraint $M$ on the joint actions, such that $\sum_{i=1}^{N} a_i \leq M$, we formulate the following integer program to determine the optimal joint action that maximizes the joint Q-network for any state $s$:

$$\begin{aligned} \underset{a^1,\ldots,a^N}{\text{maximize}} \quad & \sum_{i=1}^{N} Q'(i, s^i, a^i; \theta) \\ \text{s.t.} \quad & \sum_{i=1}^{N} a^i \leq M, \; a^i \in \mathbb{N}. \end{aligned} \tag{28}$$

This integer program, which can be efficiently solved using commercial solvers such as Google OR-Tools (Perron & Furnon, 2024) or Xpress (FICO, 2023), serves two purposes: it generates feasible data for the replay buffer to compute the expectation in (3) during training, and it determines the optimal actions once learning has converged. Notably, this optimization step is separate from the Q-learning process.

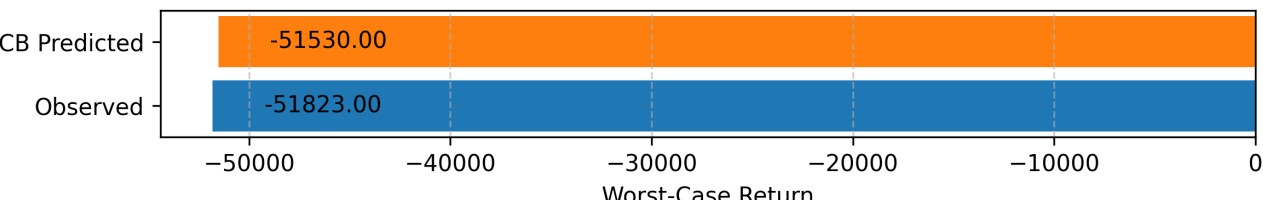

Figure 11: Comparison of worst-case returns: estimation using $Q_{\text{CB}}$ for both immediate rewards and transition probabilities is compared against the actual observed values from extensive state-action space exploration. The close alignment validates our approach of using $Q_{\text{CB}}$ to approximate both components of the worst-case scenario in (30).

## C.5. Distributionally Robust Bellman Operator in Large-Scale Robotic Sortation Warehouses

In large-scale robotic sortation warehouses, we observe that the transition probability is dependent on the induction random variable $X$, which violates the assumption in Lemma 3.2. Consequently, we must compute:

$$\inf_{\mathbb{P} \in \mathfrak{M}} \gamma \, \mathbb{E}_{p_{s,a}(X), X \sim \mathbb{P}_g} \left[ \max_{a'} \tilde{Q}(s', a') \right]$$

for the distributionally robust Bellman operator. This leads to:

$$\tilde{\mathcal{T}}_{\mathfrak{R}}(\tilde{Q})(s,a) = \inf_{g \in \mathcal{G}} \left\{ \mathbb{E}_{X \sim \mathbb{P}_g}[r(s,a;X)] \right\} + \inf_{\mathbb{P} \in \mathfrak{M}} \left\{ \gamma \, \mathbb{E}_{p_{s,a}(X), X \sim \mathbb{P}_g} \left[ \max_{a'} \tilde{Q}(s', a') \right] \right\} \tag{29}$$

where the computation becomes infinite-dimensional and practically intractable. To address this, during training, we approximate the distributionally robust Bellman operator with:

$$\tilde{\mathcal{U}}_{\mathfrak{R}}(\tilde{Q})(s,a) = \inf_{g \in \mathcal{G}} \left\{ \mathbb{E}_{X \sim \mathbb{P}_g}[r(s,a;X)] + \gamma \, \mathbb{E}_{p_{s,a}(X), X \sim \mathbb{P}_g} \left[ \max_{a'} \tilde{Q}(s', a') \right] \right\}, \tag{30}$$

which provides an upper bound for the distributionally robust Bellman operator, as shown by:

$$\begin{aligned}
\tilde{\mathcal{U}}_{\mathfrak{R}}(\tilde{Q})(s,a) &\geq \inf_{g \in \mathcal{G}} \left\{ \mathbb{E}_{X \sim \mathbb{P}_g}[r(s,a;X)] \right\} + \inf_{g \in \mathcal{G}} \left\{ \gamma \, \mathbb{E}_{p_{s,a}(X), X \sim \mathbb{P}_g} \left[ \max_{a'} \tilde{Q}(s', a') \right] \right\} \\
&\geq \inf_{g \in \mathcal{G}} \left\{ \mathbb{E}_{X \sim \mathbb{P}_g}[r(s,a;X)] \right\} + \inf_{\mathbb{P} \in \mathfrak{M}} \left\{ \gamma \, \mathbb{E}_{p_{s,a}(X), X \sim \mathbb{P}_g} \left[ \max_{a'} \tilde{Q}(s', a') \right] \right\} \\
&= \tilde{\mathcal{T}}_{\mathfrak{R}}(\tilde{Q})(s,a).
\end{aligned} \tag{31}$$

During training, we use (30) to construct the loss function (13) for DRMARL, where the optimization problem within (30) is solved using the solution from the CB-based worst-case reward estimator $Q_{\text{CB}}$. In practice, this approximation (30) proves highly effective for the worst-case return, with the relative approximation error of $\tilde{\mathcal{U}}_{\mathfrak{R}}(\tilde{Q})(s,a)$ to $\tilde{\mathcal{T}}_{\mathfrak{R}}(\tilde{Q})(s,a)$ being less than $0.57\%$ (see Figure 11). This small error margin indicates that $\tilde{\mathcal{U}}_{\mathfrak{R}}(\tilde{Q})(s,a)$ does not impede DRMARL's ability to capture the worst-case return.

## D. Additional Simulation Results and Discussions

### D.1. Simplified Sortation Environments

As shown in Table 6, the DRMARL policy with $Q_{\text{CB}}$ outperforms both the MARL policy and the DRMARL policy trained with random group selection across all metrics. Here, random group selection refers to replacing Line 7 of Algorithm 2 with:

$$g' \leftarrow \text{Uniform}(\mathcal{G}),$$

instead of:

$$g' \leftarrow \arg\min_{g \in \mathcal{G}} Q_{\text{CB}}(s_t, a_t, g).$$

This confirms that $Q_{\text{CB}}$ effectively enables the DRMARL policy to explore worst-case reward functions during training. Furthermore, when compared to a DRMARL policy trained using exhaustive search over worst-case rewards for each

Table 6: Comparison of key sortation metrics across policies, averaged over $m = 9$ groups, each evaluated over 100 runs.

| Policy | Recirculation Rate ($\downarrow$) | Throughput ($\uparrow$) | Recirculation Amount ($\downarrow$) |
|---|---|---|---|
| MARL | $2.16\% \pm 2.35\%$ | $11740.98 \pm 42.30$ | $259.02 \pm 19.81$ |
| DRMARL (random) | $1.56\% \pm 1.45\%$ | $11812.77 \pm 30.14$ | $187.23 \pm 17.26$ |
| DRMARL (with $Q_{\mathrm{CB}}$) | $\mathbf{0.56\% \pm 0.18\%}$ | $\mathbf{11932.21 \pm 24.12}$ | $\mathbf{67.79 \pm 5.16}$ |
| DRMARL (exhaustive) | $0.55\% \pm 0.13\%$ | $11933.68 \pm 22.48$ | $66.32 \pm 4.83$ |
| MARL (group-specific) | $0.53\% \pm 0.14\%$ | $11936.60 \pm 23.11$ | $63.40 \pm 5.02$ |

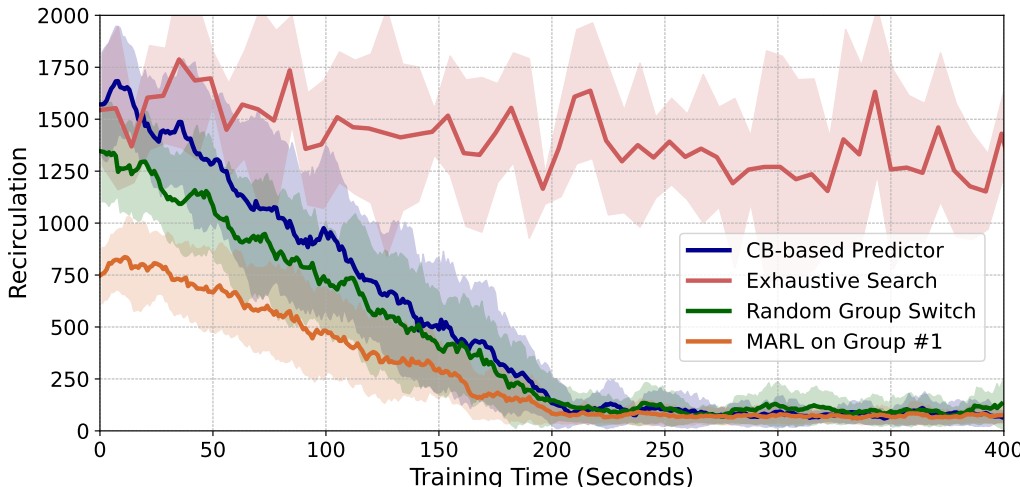

Figure 12: Training efficiency comparison in simplified robotic sortation warehouses.

state-action pair $(s, a)$, the $Q_{\mathrm{CB}}$-based approach achieves equivalent policy performance and robustness while being computationally more efficient. The last row presents the theoretical optimal performance of group-specific MARL policies (trained and tested on the same group). DRMARL performs only marginally below this optimal baseline, demonstrating an effective balance between individual performance and distributional robustness.

The training efficiency comparison across different approaches is presented in Figure 12. DRMARL with exhaustive search requires the longest training time due to its comprehensive exploration across all distribution groups. In contrast, DRMARL with $Q_{\mathrm{CB}}$ converges significantly faster while matching the recirculation performance of the exhaustive search in initial stages, validating $Q_{\mathrm{CB}}$'s ability to identify worst-case groups. DRMARL with random group selection shows faster convergence than the $Q_{\mathrm{CB}}$-based approach, but this is because random exploration does not guarantee capturing the worst-case reward functions. The group-specific MARL policy exhibits the fastest convergence due to the relative simplicity of its training task.

### D.2. Large-Scale Robotic Sortation Environments

#### D.2.1. DETAILED EVALUATION RESULTS ACROSS ALL GROUPS

Table 7 presents detailed validation results comparing MARL and DRMARL chute mapping policies across all induction distribution groups from Years 1-4. The DRMARL policy demonstrates superior performance across most groups, achieving both higher package sortation throughput and lower recirculation rates. The only exceptions are two groups in Year 4, where the MARL policy shows marginally better throughput but at the cost of higher recirculation rates. This is expected behavior since the MARL policy is specifically trained on Year 4 induction data, while DRMARL optimizes for robustness rather than throughput maximization. Overall, DRMARL achieves significant improvements, reducing recirculation by $80\%$ on average while simultaneously increasing throughput by $5.62\%$ on average.

Table 7: Relative key metrics improvements over MARL baseline trained on Year 4 data in large-scale robotic sortation warehouses.

| GROUP NUMBER | YEAR | RECIRCULATION RATE REDUCTION (%) | PACKAGE THROUGHPUT INCREASE (%) | PACKAGE RECIRCULATION AMOUNT REDUCTION (%) |
|---|---|---|---|---|
| 1 | 1 | 94.75 | 31.66 | 94.21 |
| 2 | 1 | 93.19 | 23.47 | 92.55 |
| 3 | 1 | 94.62 | 35.69 | 94.26 |
| 4 | 1 | 93.85 | 25.94 | 93.85 |
| 5 | 1 | 95.90 | 19.50 | 95.67 |
| 6 | 2 | 90.47 | 5.79 | 90.44 |
| 7 | 2 | 91.27 | 5.24 | 91.11 |
| 8 | 2 | 77.22 | 4.13 | 77.22 |
| 9 | 2 | 83.34 | 2.49 | 83.34 |
| 10 | 2 | 85.12 | 5.07 | 85.12 |
| 11 | 2 | 92.85 | 6.97 | 92.82 |
| 12 | 3 | 84.22 | 4.54 | 84.59 |
| 13 | 3 | 83.14 | 15.89 | 81.57 |
| 14 | 3 | 85.53 | 4.17 | 86.04 |
| 15 | 3 | 83.63 | 5.51 | 83.58 |
| 16 | 3 | 84.24 | 5.52 | 84.19 |
| 17 | 4 | 34.18 | -0.99 | 36.71 |
| 18 | 4 | 57.04 | -5.85 | 62.34 |
| 19 | 4 | 75.78 | 4.47 | 75.73 |
| 20 | 4 | 66.59 | 9.85 | 64.18 |
| 21 | 4 | 75.07 | 5.64 | 75.83 |

### D.2.2. ABLATION STUDY

Ablation studies were conducted across a variety of warehouse layouts, focusing primarily on varying the total number of eject chutes and unique induction destinations. These factors directly influence the action space size and the underlying transition probabilities, thereby significantly affecting policy performance. By systematically modifying these environmental parameters, we assess the adaptability and robustness of DRMARL compared to the standard MARL baseline under diverse operational conditions.

In Figure 13, we reduce the total number of eject chutes available in the warehouse to model different warehouse operational conditions. As the number of chutes decreases, competition among destinations for shared eject chute resources intensifies, amplifying the complexity of the allocation problem. Despite this increased challenge, DRMARL consistently outperforms the standard MARL baseline across all tested chute configurations. This superior performance highlights DRMARL's robust ability to handle both the complexity and uncertainty inherent in real-world resource allocation tasks.

Meanwhile, Figure 14 explores the impact of progressively reducing the number of induction destinations on policy effectiveness. DRMARL maintains stable performance throughout most scenarios, demonstrating robustness to changes in destination diversity. However, when only $50\%$ of destinations remain active, MARL surpasses DRMARL in performance. This phenomenon arises because DRMARL adopts a conservative strategy that reserves chutes for potential future destinations, including those that may not materialize when half of the destinations are inactive. In contrast, MARL employs a more aggressive allocation approach, prioritizing immediate assignment of chutes to currently active destinations without accounting for the risk of obstructing future inductions. This trade-off underscores DRMARL's cautious, risk-aware design, which generally yields better performance but can be slightly less optimal in highly reduced destination scenarios.

### D.2.3. POTENTIAL LIMITATIONS

A natural concern with DRMARL's contextual bandit-based worst-case reward estimator is whether it might induce overly pessimistic learning dynamics. Specifically, whether selecting the most adverse induction distribution at each step could lead to excessively negative reward signals, resulting in instability or premature convergence to suboptimal policies, which is a well-known challenge in robust policy learning. To mitigate such effects, our framework aggregates induction data across multiple operational days rather than evaluating each day in isolation. This aggregation dampens the influence of extreme outliers and stabilizes the worst-case signal used for training. In our experiments, this approach proved effective, as no instability was observed due to extreme worst-case distributions. Nonetheless, we acknowledge that, in practical

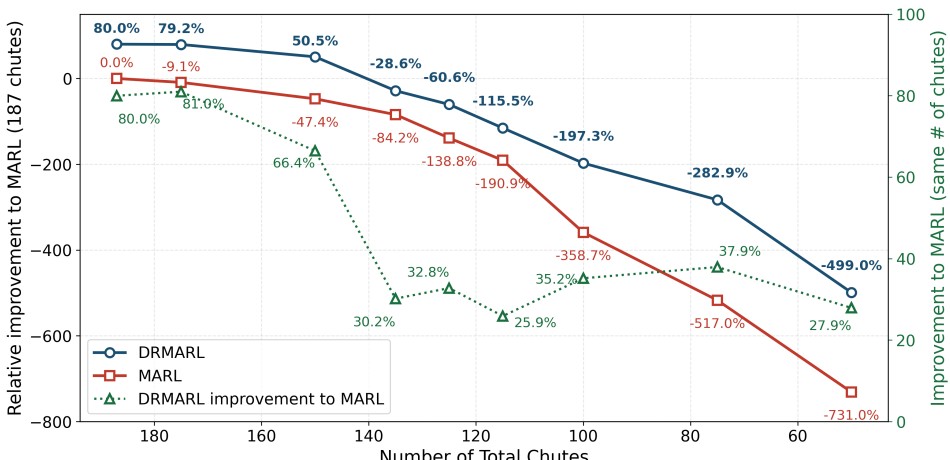

Figure 13: Relative improvement in recirculation rate (↑) over the baseline MARL across different numbers of available chutes.

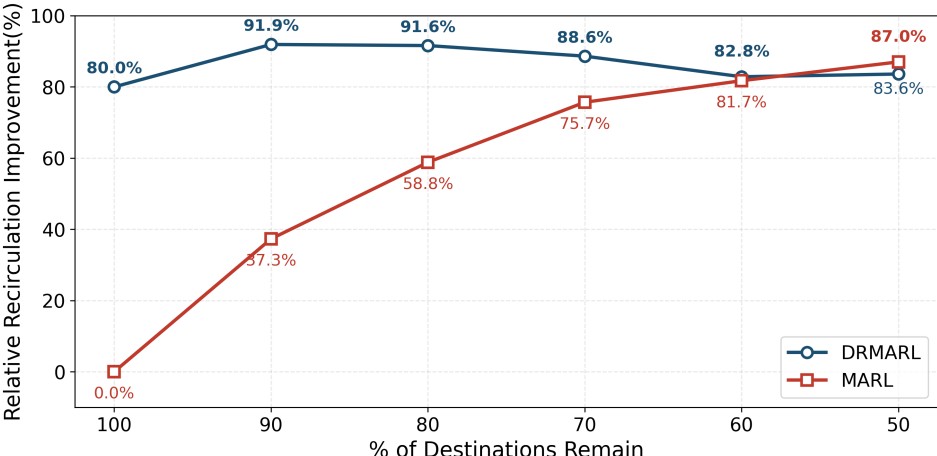

Figure 14: Relative improvement in recirculation rate (↑) compared to the baseline MARL, shown across varying percentages of 120 induction destinations remaining active.

deployment, highly anomalous induction patterns could arise. In such cases, extreme outliers can be filtered prior to training, and a fallback to a heuristic policy can be used when necessary. This trade-off reflects a known limitation of distributionally robust methods: while they offer improved resilience under uncertainty, they may become overly conservative in the face of extreme but low-probability events.

