# OpenReview forum: "Distributionally Robust Multi-Agent Reinforcement Learning for Dynamic Chute Mapping"
_ICML.cc/2025/Conference — ICML 2025 poster_

### Official Review · Reviewer_kJH7 · 2025-03-02

**Overall Recommendation:** 4

**Summary:**

This paper attempts to optimize the dynamic chute mapping problem to optimize throughput/reduce recirculation rates by formulating the problem as a multi-agent RL problem. The authors then extend the vanilla MARL framework by introducing concepts from group distributionally robust optimization into the framework and further optimize the computation efficiency of the proposed framework by using a contextual-bandit based worse-case reward predictor. Experimental results on a simple toy problem and large scale simulations shows the efficacy of the proposed method.

## Update after rebuttal

I thank the authors for addressing my comments and I find the response sufficient. As such, I will maintain my previous recommendation of accept.

**Claims And Evidence:**

Yes, claims are supported

**Essential References Not Discussed:**

N/A

**Experimental Designs Or Analyses:**

Yes, experiments are sound

**Methods And Evaluation Criteria:**

Yes, proposed method and evaluation criteria makes sense

**Other Comments Or Suggestions:**

In Eq. 1, it is not clear why is there an addition $-2a^i_t$ in the reward function

I would suggest mentioning the detail listed in the appendix of using an external tool to resolve the feasibility of joint actions in the main paper as it seems like a critical part of the framework.

Additionally, a slightly more detail explanation of the problem (dynamic chute mapping) would also improve the readability of the paper for readers not familiar with this application.

It is not clear why are the observation spaces different for the small scale problem vs the large scale problem, as listed in the appendix.

I understand that the large scale experiments are already conducted on real world data, but as always, additional experiments on different simulation parameters, for example, varying the number of destination points/number of chutes and demonstrating that the conclusions still hold would make the paper stronger.

**Other Strengths And Weaknesses:**

Strengths:

Overall, I believe this is a strong paper. The paper is well-written, organized and structured. The idea of applying Group DRO concepts to MARL and the introduction of a contextual bandit-based reward predictor to improve efficiency is novel to the best of my knowledge. The experimental results also supports the claims with sufficient theoretical justification and I believe the method introduced in this paper may also translate beyond the application of dynamic chute mapping.

Weakness:

Certain parts of the paper could be better organized in order to improve the readability for reader either not familiar with RL/Dynamic Chute mapping. The paper is also lacking of additional experiments and ablation studies for different parameters of the environment.

**Questions For Authors:**

1. Given that the contextual bandit-based worst-case reward predictor selects the most adverse induction distribution at each step, have the authors observed scenarios where the worst-case reward signal is excessively negative, leading to instability in learning dynamics or premature convergence to a suboptimal policy? Additionally, how does the framework mitigate the risk of overly pessimistic policy updates that may arise from extreme worst-case scenarios?

2. The paper compares the proposed method primarily against MARL and exhaustive search, but how does it perform relative to other robust RL approaches, such as adversarial training methods that explicitly perturb the environment to simulate worst-case scenarios?

3. Are there any foreseeable drawbacks of the proposed method over existing methods?

**Relation To Broader Scientific Literature:**

I believe this paper is generally related to the area of robust MARL as well as relevant to the areas of operation research.

**Theoretical Claims:**

Yes, the proofs of Lemma 3.1 and 3.2

---

> ### Author Rebuttal · Authors · 2025-03-31
>
> Thank you for your insightful comments, thoughtful questions, and encouraging feedback. Your suggestions greatly improve the paper. Below are our responses, following the order of the reviewer comments, with references to tables and papers prefixed by “R-” for clarity.
>
> We address all readability and presentation concerns in the revision.
>
> In (1), actions are introduced as penalty to prevent MARL from over-allocating chutes to a few destinations, which could block incoming packages from other destinations. However, MARL still fails to generalize to OOD induction patterns. We will clarify this in the revision.
>
> For small-scale problems, the observation space excludes recirculated packages, making the transition probability independent of induction $X$ and aligning with the assumptions in Lemma 3.2. For large-scale problems, as shown in Appendix C.5, the DR Bellman operator in Lemma 3.2 acts as an accurate estimator when the transition probability depends on $X$, which does not hinder DRMARL from learning an effective robust policy.
>
> Ablation studies were performed under various warehouse layouts, primarily varying the number of chutes and unique induction destinations, which directly affect the action space and transition probability. STDs omitted due to character limits.
>
> |# of Chutes|50|100|115|125|135|150|187|
> |-|-|-|-|-|-|-|-|
> |DRMARL|**-499.00%**|**-197.33%**|**-115.54%**|**-60.59%**|**-28.64%**|**50.52%**|**79.97%**|
> |MARL|-730.97%|-358.70%|-190.91%|-138.85%|-84.25%|-47.42%|0% (baseline)|
>
> **TABLE R-3**: Relative recirculation rate improvement (↑) compared to the baseline MARL.
>
> |% of Destinations|50|60|70| 80|90|100 (max)|
> |-|-|-|-|-|-|-|
> | DRMARL| 83.58% | **82.82%** | **88.60%** | **91.57%** | **91.88%** | **79.97%** |
> | MARL| **86.99%** | 81.71% | 75.66% | 58.81% | 37.31% | 0% (baseline) |
>
> **TABLE R-4**: Relative recirculation rate improvement (↑) compared to the baseline MARL across different percentages of 120 destinations remains in the induction.
>
> In Table R-3, DRMARL consistently outperforms MARL, demonstrating its effectiveness and robustness across different environments. In Table R-4, we examined the performance of both policies while gradually reducing the number of induction destinations. DRMARL maintained its performance, but MARL outperformed it when only 50% destinations remained. This occurs because DRMARL conservatively reserves available chutes for potential future inducts, even for destinations that have not yet appeared and may never appear with 50% destinations. In contrast, MARL aggressively allocates chutes to existing destinations without considering the risk of blocking newly inducted destinations.
>
> Overly pessimistic decisions is a common risk in robust policies. We mitigate this by grouping induction data from multiple days rather than treating each day separately, reducing the impact of extreme outliers. Our experiments did not exhibit instability from extreme worst cases. However, if such cases arise, one can remove extreme outliers to stabilize training and switch to a heuristic policy in practice when extreme induction patterns are detected. This is a foreseeable drawback compared to MARL since the policy may become overly conservative with extreme inductions.
>
> We compare DRMARL with 3 Robust RL (RRL) approaches: <1> an adversarial agent deliberately blocking available chutes [R-4], <2> an adversarial environment with perturbed transition probabilities [Pinto et al., 2017], and <3> a (non-distributionally) robustified reward function [Wiesemann et al., 2013]. Implementation and result details are provided in the revision. While certain robust policy (<2>) achieve comparable cumulative recirculation performance to DRMARL, DRMARL additionally guarantees improved worst-case performance by explicitly optimizing for it. To validate this, we analyze the empirical distributions of recirculated packages across all time steps and groups. In Table R-5, we report the relative reduction in both the worst-case and CVaR (5% confidence level) of recirculated packages. DRMARL outperforms all RRL due to its distribution awareness, leading to more accurate worst-case reward estimation and robust action value functions, ultimately optimizing worst-case performance. This is crucial, as sudden surges in recirculation can cause robot congestion, delaying the entire sorting process. Since congestion is not explicitly modeled in the simulation, DRMARL’s advantage over RRL is expected to be even greater in practice.
>
> |Method|CVaR reduction(↑)|Worst-case recirc reduction(↑)|Recirc rate reduction (↑)|
> |-|-|-|-|
> |<1>|61.11%|53.57%|66.19%|
> |<2>| 58.48%|23.68%|77.09%|
> |<3>|58.97%|60.23%|70.43|
> |DRMARL|**77.09%**|**76.61%**|**79.97%**|
> **TABLE R-5**: Relative step-wise worst-case recirc amount, CVaR, and recirc rate reduction compared to baseline MARL.
>
> **References:**
>
> [R-4] Mandlekar et al., Adversarially Robust Policy Learning: Active construction of physically-plausible perturbations, IROS 2017.

---

### Official Review · Reviewer_iKz7 · 2025-03-12

**Overall Recommendation:** 2

**Summary:**

The paper addresses the “dynamic chute mapping” task in robotic warehouses, where packages must be assigned to chutes in the face of uncertain and shifting arrival (induction) patterns. It proposes a distributionally robust multi-agent reinforcement learning (DRMARL) framework, combining group distributionally robust optimization with MARL to prepare for worst-case variations in package flow. The authors introduce a contextual bandit–based predictor that selects the likely worst-case distribution group for each state-action pair, reducing computational overhead compared to exhaustively checking every group. They demonstrate in both a simplified and a large-scale warehouse simulator that their DRMARL approach consistently lowers package “recirculation” (re-sorting) and achieves improved throughput across a range of induction scenarios, including out-of-distribution ones.

**Claims And Evidence:**

The paper applies a group DRO (distributionally robust optimization) approach to multi-agent reinforcement learning (MARL) for a warehouse “chute mapping” problem. Their claim is that, by anticipating worst-case changes in package arrival rates, the learned policy is more “robust” than a regular MARL policy. They present some experiments in both small and large simulation settings that show lower “recirculation” (which is re-sorting effort) and better throughput when induction rates deviate from the usual pattern. The evidence is mainly the simulation results comparing their DRMARL approach to standard MARL or simpler baselines.

While the simulation outcomes look decent, the evidence is somewhat specialized to their chute mapping scenario. They also have some basic group DRO math (like rewriting the Bellman operator) to explain how they handle uncertain reward distributions. But it’s not a deep theoretical contribution—more of an application.

**Essential References Not Discussed:**

Nothing noteworthy to me.

**Experimental Designs Or Analyses:**

The main experiment is to train a policy on multiple “groups” (induction patterns), then test on new patterns. They show that their approach has better worst-case performance than normal MARL. This setup is straightforward and in line with standard RL or robust RL experiments.

The chute system is a bit niche, so it’s unclear if this generalizes to bigger or different warehouse tasks. Also, they only compare to baseline MARL or a naive robust policy, so we don’t see how it might stack up against other specialized robust RL frameworks. The experiments seem valid but narrow.

**Methods And Evaluation Criteria:**

They start from a standard MARL setting, add group DRO for different induction patterns, and then do a contextual bandit trick to pick the “worst-case” induction group. They evaluate performance by how many packages get recirculated, how many total packages get sorted (“throughput”), and how stable the policy is across multiple induction distributions.

The method is essentially an existing robust RL idea (DRO + MARL) applied to their warehouse environment. Their evaluation metrics make sense in that domain (recirculation rate, throughput). But it doesn’t push the boundaries for new RL or new optimization ideas. It’s more about showing that existing robust RL methods can handle uncertain arrival distributions.

**Other Comments Or Suggestions:**

It would help if you provided concrete evidence or references showing that real warehouses do experience large swings in induction rates, especially since many readers at ICML may not be as familiar with that domain. Could you cite any operational research studies, internal data, or industrial reports that quantify these variations and explain why they're significant enough to justify a robust RL approach?

**Other Strengths And Weaknesses:**

Strength:
1. The paper tests an actual warehouse scenario, which is relevant to industry.

Weakness:
1. Conceptually, it’s mostly applying known ideas (group DRO, multi-agent RL, contextual bandits) rather than introducing a fundamentally new theory.
2. The chute problem is a narrow application, so broader impact could be limited unless it’s widely adopted in automated warehouses.
3. It’s not fully clear if the “worst-case distribution group” approach is truly necessary if the environment doesn’t vary that much in practice.

**Questions For Authors:**

1. Do you see a path for applying the same idea to other resource-allocation tasks beyond warehouse sorting? If so, have you tested on simpler domains?
2. Could a standard robust RL approach (without group DRO or contextual bandit) handle moderate variability in induction rates almost as well? It’d be helpful to see a direct comparison.

**Relation To Broader Scientific Literature:**

They cite work in robust RL, distributionally robust optimization, and multi-agent resource allocation, and they connect it to existing RL-based sorting approaches (like the MARL method from Shen et al. 2023). The main novelty is combining group DRO and multi-agent RL with a “contextual bandit” step for computational speed. But conceptually, these are known techniques—there’s not a big new theory angle here. They haven’t proven new theorems that break ground on robust MARL.

**Theoretical Claims:**

The paper does not have theoretical contribution.

---

> ### Author Rebuttal · Authors · 2025-03-30
>
> Thank you for your insightful comments and thoughtful questions.  Your suggestions greatly improve the paper. Below are our responses, following the order of the reviewer comments, with references to tables prefixed by “R-” for clarity. If any referred content is missing here, please find it in the responses to other reviewers.
>
> The proposed framework builds on existing methods, but their integration is novel and strategically bridges gaps in DRRL, particularly related to scalability for large-scale multi-agent RL problems. In the robotic sortation warehouse, the reward (recirculation) function is implicit and highly nonlinear. Consequently, unlike typical well-structured DRO formulations with finite-dimensional convex equivalent formulations, applying similar methods to the complex, large-scale chute mapping problem is infeasible. With millions of packages sorted daily, estimating the worst-case reward for each state-action pair becomes computationally impractical using existing DRRL techniques. Although group-DRO makes worst-case reward evaluation finite-dimensional, the complexity modeling of the warehouse sortation and the large number of groups render exhaustive search still infeasible. Unlike supervised learning, where soft reweighting can avoid exhaustive search, such techniques cannot be directly applied to RL, as the worst-case group and reward vary across state-action pairs. An important contribution of our work is to leverage a contextual bandit to significantly reduce the exploration cost for estimating the reward of the worst case group. This approach effectively bridges the gap in **scalability** and **applicability**.
>
> We conducted extensive experiments for the comparison between DRMARL and naive Robust RL. Please kindly find the detailed result in Table R-5. Compared to naive Robust RL, DRMARL benefits from its distribution awareness for more accurate estimation of the worst-case reward and robust action value functions. Even certain naive Robust RL policy (<2>) achieves a comparable cumulative recirculation throughout the day, DRMARL still significantly outperforms it in terms of the step-wise recirculation amount (both worst-case and CVaR), which is the direct metric for robustness when comparing the performance of robust policies.
>
> The proposed DRMARL framework is designed to address general RL problems where the reward function undergoes distribution shifts. Since Lemmas 3.1 and 3.2 are not specifically tailored to resource allocation problems, the framework is broadly applicable to other DRRL settings. It can also benefit large-scale RL problems with group-DRO, particularly when an exhaustive search is impractical. We see strong potential for extending this approach beyond sorting problems, as DRMARL directly addresses reward function ambiguity (i.e., environmental uncertainty). The core DRMARL framework (Algorithms 1 and 2) remains unchanged when applied to other resource allocation problems, with only the environment varying. We are currently working on extending this framework to general RL problems beyond resource allocation, with early-stage promising results, and we will report our findings in a separate manuscript.
>
> Major e-commerce companies have publicly reported double digit growth in retail sales YoY (Year-over-Year) in their Q4 earnings press releases on multiple years which translates in the package sortation context to large swings in induction rates YoY. Due to the double-blind policy, we will provide reference in the final paper. Package volume and distribution exhibit substantial seasonal and yearly shifts. Seasonality drives fluctuations during sales events and holidays, while yearly shifts reflect evolving customer behavior and business growth. To support our claims, we quantify distribution changes using the empirical Type-1 Wasserstein Distance [R-3], as shown in Table R-2. By comparing four selected weeks to Week 1 within each year, we reveal significant seasonal variations, while the Wasserstein distance between consecutive days is typically on the order of 20. The last column highlights yearly shifts by comparing the same period across years to Year 1, showing that yearly changes are even more pronounced than seasonal ones. Figure 5 and Table R-1, evaluated on production data, demonstrate that MARL struggles with large induction pattern changes, underscoring the necessity of DRMARL.
>
> |Week #|2|3|4|5|Avg Dist to Year 1|
> |-|-|-|-|-|-|
> |Year 1|76.87|202.96|147.17|241.80|0.0|
> |Year 2|36.46| 70.47|301.25|123.45|1172.85|
> |Year 3|25.81| 23.87|72.91|78.82|585.98|
> |Year 4|72.99| 45.04|24.92|62.39|288.86|
>
> **Table R-2**: The first four columns represent Type-1 Wasserstein distances for each week compared to Week 1 of the same year. The last column represents the average Wasserstein distance of each year to Year 1.
>
> **References:**
> [R-3] Villani, Cédric. Optimal transport: old and new. Vol. 338. Berlin: springer, 2008.

---

### Official Review · Reviewer_AFKN · 2025-03-13

**Overall Recommendation:** 3

**Summary:**

This paper proposes the Distributionally Robust Multi-Agent Reinforcement Learning (DRMARL) framework for dynamic chute mapping in robotic warehouses. The integration of group Distributionally Robust Optimization (DRO) with a contextual bandit-based predictor to handle induction rate variations is the main contribution. However, there are some presentation issues for this paper. I will consider increase the score if the authors can make proper improvement.

**Claims And Evidence:**

This paper proposes DRMARL algorithm which is validated in simulated and large-scale environment.

**Essential References Not Discussed:**

The paper provides extended literature reviews.

**Experimental Designs Or Analyses:**

The experiments are reasonable.

**Methods And Evaluation Criteria:**

The proposed algorithm makes sense.

**Other Comments Or Suggestions:**

•	Overall, the presentation is not very clear in several places.
•	For the theory part, the two lemmas provided are not very strong. Could the author provide, e.g., a convergence analysis/the contraction property of the distributionally robust Bellman operator
•	Induction rate is the key concept of this paper, but the paper never gives a clear description of it. This makes the paper inaccessible to the audiences of ICML. Is it a real number or does it contain other information?
•	A similar problem applies to DRO.
•	What are the features of the observation? What are the differences between state and observation in this chute mapping problem since the paper models both state and observation in L162.
•	L176 refers to Figure 5 that appears several pages after the description.
•	For Lemma 3.2, is it a definition or a theoretical result? What is the definition of distributionally robust Bellman operator? What is the benefit of using it in the algorithm?
•	How many times do you run the policies for each group in Table 1? Why do the authors not show the std of throughput and recirculation amount?
•	The notation X appears in L372 seems to conflict with the notation $X$ that appears in e.g., Eq. 11~13.
•	The authors define the ambiguity set as a convex combination of past patterns. Could this modeling capture the patterns that gradually change year by year (e.g., the more and more packages for promoting seasoning of each year)? The authors try to address this in experiment results shown in Figure 6. But how far is the test distribution from the ambiguity set?
•	In Figure 9, why the rewards decrease along with training?
•	For Figure 5, are Year 1-4 sorted from past to current? A natural test would be training the model on Year 1 and test it on the later years. Why do the authors reverse?
•	Form the statistical values, DRMARL improves significantly over the MARL baseline. But could the authors provide detailed cases where the DRMARL policy outperforms the baseline? E.g., the strategies of it in extremely busy scenarios?

**Other Strengths And Weaknesses:**

Strengths
•	The proposed algorithm that combines group DRO with MARL is novel and reasonable.
•	The literature review is extended and well organized.
•	The experiment results are comprehensive.
•	The experiment shows significant improvement over the baseline MARL algorithm.

See weaknesses in the below part.

**Questions For Authors:**

See the above

**Relation To Broader Scientific Literature:**

The paper proposes a new method for the chute mapping tasks in the field of operation research. This method has some potential to be applied to broader fields.

**Theoretical Claims:**

The theoretical part is weak, but the current statements are reasonable.

---

> ### Author Rebuttal · Authors · 2025-03-30
>
> Thank you for your insightful comments, thoughtful questions, and encouraging feedback. Your suggestions greatly improve the paper. Below are our responses, following the order of the comments, with references to tables and papers prefixed by “R-” for clarity. If any referred content is missing here, please find it in the responses to other reviewers.
>
> The contraction property of the DR Bellman operator is shown in the proof of Lemma 3.2 (Line 668). We explicitly highlight this in the revision for clarity, and replace the paragraph starting from Line 681 with:
>
> Since $\gamma \in [0,1]$, this establishes that DR Bellman operator is a contraction mapping under the $\ell\infty$ norm. By Banach’s Fixed Point Theorem [R-1], there exists a unique fixed point $\tilde{Q}^*$ such that $\tilde{\mathcal{T}}_{\mathcal{G}}(\tilde{Q}^*) = \tilde{Q}^*$. Consequently, iteratively applying the operator ensures convergence to $\tilde{Q}^*$, proving the stability of the robust Q-learning algorithm. This contraction property of the DR Bellman operator was also addressed in [R-2] when the ambiguity set is defined for transition probability.
>
> Due to the character limit, the induction rate, in short, is a vector that contains number (integer) of packages inducted per destination and hour. A comprehensive definition of induction rate/pattern is provided in the revision.
>
> We provide a more detailed explanation of DRO in the revision.
>
> In this formulation, the state space $\mathcal{S}$ represents the global state, while each agent $i$ has a local observation $\mathcal{O}_i \subset \mathcal{S}$ containing partial information about the global state. The observation features are detailed in Appendix C.1, Line 739. While the state provides full system information, each agent’s observation is limited to local aspects relevant to its decision-making. This partial observability requires agents to act based on their own experiences and available information. We will refine our explanation in Line 162 to better highlight this distinction.
>
> We place the Figure 5 near Line 176 in the revision.
>
> Lemma 3.2 defines the DR Bellman operator, derives its explicit form for group DRO with MARL, and establishes its contraction property. Its formal definition appears in Equation (20), Line 655, with benefits discussed in Line 220. Notably, minimizing the worst-case Bellman error among groups does not necessarily yield a policy optimal under worst-case rewards. This is because the worst-case Bellman error reflects the worst-case deviation from the target Q-function but does not guarantee convergence to the optimal robust Q-function. Instead, the DR Bellman operator and its corresponding DR Bellman error ensure robustness. We refine Lemma 3.2 and the corresponding text for clarity in the revision.
>
> In Table 1, each group is tested 100 times. STDs are omitted for table readability, and are included in the revision.
>
> In Line 372, $X$ denotes the induction pattern (packages per destination per hour) and is a realization of a random variable following the induction generating distribution. Thus, the immediate reward function (recirculation) depends on $X$.
>
> Convex combinations of distribution groups aim to span the space of potential induction distributions, using available distribution groups to model the space of potential target distributions is a common approach in DRO. As we collect more production data, the distribution groups will expand to better capture year-to-year shifts without increasing the DRMARL training complexity. Figure 6 shows that the robust policy generalizes well to test distributions specifically designed outside $\mathfrak{M}$ without theoretical guarantees. Its Type-1 Wasserstein distance to $\mathfrak{M}$ is 818.19, while the average distance among distributions within $\mathfrak{M}$ is 542.96.
>
> In Figure 9, the reward is indeed increasing as the y-axis is flipped. Fixed in the revision.
>
> The years are ordered from past to present. Year 4 was chosen arbitrarily. In Table R-1, training the MARL policy on Year 1 also results in significant performance degradation on OOD data, with similar outcomes as MARL trained on Year 4 when compared with DRMARL.
>
> |Year|2|3|4|
> |-|-|-|-|
> |Recirc Degradation(↓)|51.01% ± 0.17%|74.45% ± 0.22%|42.89% ± 0.64%|
>
> **Table R-1**: Relative recirculation rate degradation compared to the baseline MARL (trained on Year 1) on induction data from Year 2-3.
>
> The detailed chute allocation action and strategy difference are omitted due to the character limit and are presented in the revision. Please kindly refer to the text under Table R-4 and Table R-5 for high-level strategy difference and step-wise recirculation outcomes between MARL and DRMARL.
>
> **References:**
>
> [R-1] Rudin, W. Principles of Mathematical Analysis (3rd ed.). McGraw-Hill, 1976.
>
> [R-2] Iyengar, G. Robust dynamic programming. Mathematics of Operations Research, 2005.

---

### Decision · Program_Chairs · 2025-05-01

**Decision:**

Accept (poster)

**Comment:**

This paper proposes to use distributional robust multi-agent reinforcement learning for solving the destination-to-chute mapping problem in package sorting where there is uncertainty over the package induction rates.

This paper was somewhat borderline. All reviewers appreciated the fact that this was an application-driven paper with an interesting and non-trivial application, and that the algorithm was well suited to the problem. Nevertheless, there were some concerns around the theoretical/algorithmic contributions with some reviewers pointing out the lack of theoretical results. Overall, however I believe that this paper is a good application-driven paper and so recommend acceptance.